# Single-cell alternative polyadenylation analysis reveals mechanistic insights of COVID-19-associated neurological and psychiatric effects

Qun Chen[1,2☯], Ying Gu[1☯], Shuai Liu[1☯], Xingyu Li[1], Ruizhi Xu[1], Ruixi Ye[1], Jingjing Yang[3*], Wanshan Ning[1*]

**1** Institute for Clinical Medical Research, The First Affiliated Hospital of Xiamen University, School of Medicine, Xiamen University, Xiamen, Fujian, China, **2** Xiamen Cell Therapy Research Center, The First Affiliated Hospital of Xiamen University, School of Medicine, Xiamen University, Xiamen, Fujian, China, **3** Department of Pulmonary and Critical Care Medicine, the First Affiliated Hospital of Xiamen University, School of Medicine, Xiamen University, Xiamen, Fujian, China

☯ These authors contributed equally.
* ningwanshan@xmu.edu.cn (WN); Jingjingyang_xmu@126.com (JY)

## Abstract

COVID-19 is associated with increased risks of neurological and psychiatric sequelae. Alternative polyadenylation (APA) is ubiquitous in human genes, resulting in mRNA diversity, and has been validated to play a pivotal regulatory role in the onset and progression of a variety of diseases, including viral infections. Here, we analyzed the APA usage across different cell types in frontal cortex cells from non-viral control group and COVID-19 patients, and identified functionally related APA events in COVID-19. According to our study, the poly(A) site (PAS) usage is different among cell types and following SARS-COV-2 infection. Moreover, we found the genes with significant PAS level changes affected pathways related to RNA splicing, and neuronal development and function, suggesting that survivors of COVID-19 will have a high risk of these diseases and that alternative splicing functions cause these changes. Additionally, APA usage and its correlation with gene expression levels varied across genes, some prefer short isoform that is more stable to produce more proteins, while others may be regulated by different mechanisms. A total of 267 risk genes targeted by microRNAs for common neurological and psychiatric disorders were found to undergo significant changes in APA following infection. In conclusion, our comprehensive analysis of APA in neural cells from COVID-19 patients at the single-cell level elucidated changes in APA levels in the brains of SARS-COV-2-infected patients and confirmed that these changes impair the function of the nervous system, providing important insights for COVID-19-associated sequelae.

**Data availability statement:** The publicly available datasets used in this study can be found in GSE159812 (https://www.ncbi.nlm.nih.gov/geo/query/acc.cgi?acc=GSE159812). The data supporting the results in this study are available within the paper and its Supplementary Information. All source datasets are archived at https://www.jianguoyun.com/p/DcmMT-cQq8KtDBi_sI8GIAA. All source codes for the data analysis or figure creation are available at https://github.com/yinggu94/APA.

**Funding:** This work was supported by the National Key Research and Development Program of China (2022YFC2704300 and 2021ZD0201300), the National Natural Science Foundation of China (32400532), the Fujian Provincial Health Technology Project (2024GGB18), the Natural Science Foundation of Fujian Province, China (Grant No.2025J08313), the Fujian Science and Technology Program Guiding Project (2025D022), the China Postdoctoral Science Foundation (2021M701337 and 2022T150242), and the Project of Xiamen Cell Therapy Research Center, Xiamen, Fujian, China (3502Z20214001). The funders had no role in study design, data collection and analysis, decision to publish, or preparation of the manuscript.

**Competing interests:** The authors have declared that no competing interests exist.

## Introduction

Brain and peripheral nerve tissues express a larger proportion of the genome than any other tissue, providing a broad transcriptomic repertoire [1]. This broad expression profile, together with extensive post-transcriptional regulation-including alternative splicing, alternative transcription initiation, and alternative polyadenylation (APA)-underlies the exceptionally high mRNA diversity observed in the nervous system [2].

Polyadenylation is an essential post-transcriptional process in which precursor mRNA is cleaved at a specific poly(A) site (PAS), followed by the addition of an untemplated stretch of adenosines at the 3' end. The selective use of alternative PAS within a single gene, known as APA, generates transcript isoforms with different 3' ends and regulatory properties.

APA, which occurs in more than 70% of human protein coding genes, has recently been recognized as a key regulator of gene expression during disease onset and progression [3]. Compared with other tissues, nervous-system-expressed genes tend to have longer 3'-UTRs on average [4]. Within the nervous system, the hippocampus exhibits the greatest number and expression levels of extended 3'-UTRs [1], consistent with its critical role in memory and spatial navigation. Hippocampal injury caused by cerebral hypoxia and encephalitis can result in memory loss, and studies in mouse models of status epilepticus and epilepsy have shown that more than 25% of the transcriptome undergoes changes in poly(A) tail length [5].

COVID-19 survivors are at increased risk of developing long-term neurological disorders. The study found that 14% of the patients are detected with new cerebral ischemic lesions and 86% of patients have astrocyte proliferation, microglia activation and cytotoxic T lymphocyte infiltration in the brainstem and cerebellum [6]. In addition, COVID-19 is associated with a severe innate immune response and systemic inflammation to promote cognitive decline and neurodegenerative diseases, which makes COVID-19 survivors likely to suffer from neurodegenerative diseases in the next few years [7]. Studies have shown that polyadenylation plays an important role in the behavior of the immune system. For instance, the inhibition of polyadenylation reduces the induction of inflammatory genes, and translation is inhibited by regulating the cellular polyadenylation binding protein in inflammatory response, emphasizing the necessity of studying the regulation of polyadenylation in the immune system [8–10]. The main resident immune cells of the brain are microglia, a large number of perivascular macrophages and dendritic cells, and detectable numbers of T cells, B cells, and natural killer (NK) cells. In addition, the average 3'-UTR length shortens after vesicular stomatitis virus infection in macrophages [11].

To the best of our knowledge, the APA regulation in association with COVID-19 has not been studied in brain yet. In view of the extensive regulatory role of APA in neurons and immune response, the study of APA level in COVID-19 patients and survivors' brain tissue is helpful to understand the pathogenic mechanism of SARS-CoV-2 from the level of post transcriptional modification, which is of great significance to the nerve injury in COVID-19 patients.

## Materials and methods

### Data pre-processing and snRNA-seq quality control

The dataset is publicly available in the GEO repository (accession number GSE159812), generated using the 3' tag-based single-cell RNA sequencing protocol (10x Genomics v3) [12]. A total of 15 frontal cortex samples were selected from individuals in the COVID-19 and non-viral control groups. Donor information, including age, sex, group, and number of nuclei, is summarized in Supplementary Table 10.5 in S10 Table, with complete metadata available in the original publication. Raw gene counts were obtained by aligning sequencing reads to the hg38 genome (refdata-gex-GRCh38-2020-A) using CellRanger software (V6.0, 10x Genomics). The cells with high ratio of mitochondrial (more han 10%) and cells with <200 detected features were removed in Seurat 4.0.5 [13]. And technical metrics including median/mean UMI per nucleus and median/mean genes per nucleus are summarized in Supplementary Table 10.7 in S10 Table.

### Cells integration and annotations

After quality control, the data were further normalized, scaled, and filtered for highly variable features using Seurat functions NormalizeData, ScaleData and FindVariableFeartures prior to principal component analysis (PCA). Then 15 samples are integrated by Harmony method to reduce the individual difference. The entire dataset is projected onto two-dimensional space by UMAP (Uniform manifold approximation and projection) on the top 20 principal components following the tutorial of Seurat. We annotate cell types referring to the original paper [12] at gene expression levels.

### Gene set enrichment analysis

We use irGSEA(v.4.0.0) to analyze and comprehensively evaluate the differential gene sets for cell subsets in the control group and the COVID-19 group. Firstly, we scored individual cells by multiple gene set enrichment methods, and generated multiple gene set enrichment score matrix. Next, we calculated the differentially expressed gene sets for each cell subpopulation in the enrichment score matrix of each gene set by wilcox test. Then the robust rank aggregation (RRA) algorithm in RobustRankAggreg package is used to comprehensively evaluate the results of difference analysis, and the gene sets that are significantly enriched in most of the gene set enrichment analysis methods are screened, visualized and analyzed.

### Cells annotations at APA level

According to the tutorial from https://github.com/YangLab/SCAPTURE [14], after the raw data processing following the method described in chapter 3, we get the matrix with PAS transcripts files as the input for single-cell APA analysis, (the process is consistent with single cell gene expression for filtering outliers, normalization, integration, and clustering). The clusters are annotated with the expression of the gene transcripts with specific PAS. We processed 3' tag-based reads with SCAPTURE to call PAS, remove internal-priming artifacts via its deep-learning classifier, and generate a per-sample PAS-by-cell UMI count matrix. PAS identifiers and cell barcodes were harmonized across samples; lowly expressed PAS were filtered (present in ≥0.5% of cells per sample). The resulting PAS matrices were merged into a Seurat object and underwent the same normalization, integration (Harmony), dimensionality reduction, and clustering procedures as gene expression. For defining cell-type-specific PAS, we first calculated the mean CPM-normalized expression of each PAS within each major cell type in the control group. PAS with a mean expression > 0 in exactly one cell type and 0 in all other cell types were classified as cell-type-specific for that cell type, whereas PAS expressed in more than one cell type were classified as non-cell-type-specific.

### APA usage calculation

We quantified per-cell proximal PAS usage using the Proximal Usage Index (PUI). For each gene with ≥2 exonic PAS, $\text{PUI\_proximal} = \text{CPM\_proximal} / \Sigma(\text{CPM of all PAS for that gene})$, yielding values in [0, 1]; a higher PUI indicates

preferential proximal usage. Per-cell APA preference was summarized as the mean PUI across expressed genes. Within each gene (strand-aware), PAS were ordered by genomic position and labeled as Proximal (most 5'), Distal (most 3'), Middle (any intermediates), or Single (only one PAS); we did not use a fixed base-pair distance threshold. For binary contrasts, 'non-proximal' denotes the union of Middle + Distal, whereas 'distal' refers exclusively to the most 3' PAS. All transcripts with at least one proximal and one distal PAS site annotated in the database were included in the analysis; thus, transcripts with only a single PAS were excluded.

Cell-type PAS expression was computed from the SCAPTURE PAS-by-cell UMI matrix after per-cell CPM normalization. For each annotated cell type (Idents in Seurat), we calculated the mean CPM per PAS across cells (AverageExpression). Group comparisons (COVID-19 vs. control within the same cell type) were performed on per-cell CPM using FindMarkers (Wilcoxon rank-sum test) to obtain avg $\log_2$ fold-change and adjusted P values; significance was defined as adjusted $P < 0.05$ and $|avg \log_2 FC| > 0.25$.

### The GO analysis and KEGG analysis based on the APA

All the subtypes of cell clusters are extracted from the APA usage matrix, and the differential expressions are calculated. Transcripts with adjusted p values less than 0.05 and an average log2 fold change larger than 0.25 are selected after the data is obtained for differential analysis. The multiple transcripts with distinct PAS transcripts could be produced from a single-gene locus, and we use the genes with significant changes of PAS transcripts as the input for Gene Ontology (GO) biological function and Kyoto Encyclopedia of Genes and Genomes (KEGG) pathway. GO process and pathway enrichment analysis of genes with significant alterations of PAS were performed by metascape [15]. The similarity score between terms was determined by calculating the Kappa-test scores between each of the two representative terms chosen from the Metascape results (P value ≤ 0.05). Cytoscape [16] was employed to visualize the representative GO terms, and terms were set as nodes, and similarity scores of greater than 0.3 were set as edges.

### The validation of PAS usage by IGV

Reads from each cell type are extracted from the bam file, and the bam file is converted to the bigwig (bw) format file using deeptools. The metadata bed file from SCAPTURE is used as the reference and imported to the IGV software, and the interested cell types of bw file are imported to the IGV for visualization.

### The 3'-UTR changes in APA

The PAS located in 3'-UTR are extracted, and the corresponding transcripts were analyzed using the movAPA [17] software following its tutorial to calculate 3'-UTR changes. For each gene, movAPA calculated the ratio of PAS transcript counts in the COVID-19 group and the non-viral control group at proximal and distal sites. Differential PAS usage was defined as the usage value in the infected group minus that in the control group, with an absolute difference greater than 0.1 considered a significant change. These APA usage changes were then combined with differential gene expression results to classify genes into four categories.

## Results

### Global APA levels in neural cells

The PAS were annotated using the latest genome annotation, with 27.9% located in 3' UTRs, and the remainder distributed across other genomic regions (i.e., 5' UTR, coding sequence (CDS), and intron) (Fig 1A). Although 3' UTR PAS are not the majority, they are of particular biological interest because APA events in this region can alter mRNA stability, localization, and translational efficiency by modulating regulatory elements such as microRNA target sites and RNA-binding protein motifs. The poly(A) signals, typically located 15–30 nt upstream of the PAS, are necessary for pre-mRNA cleavage

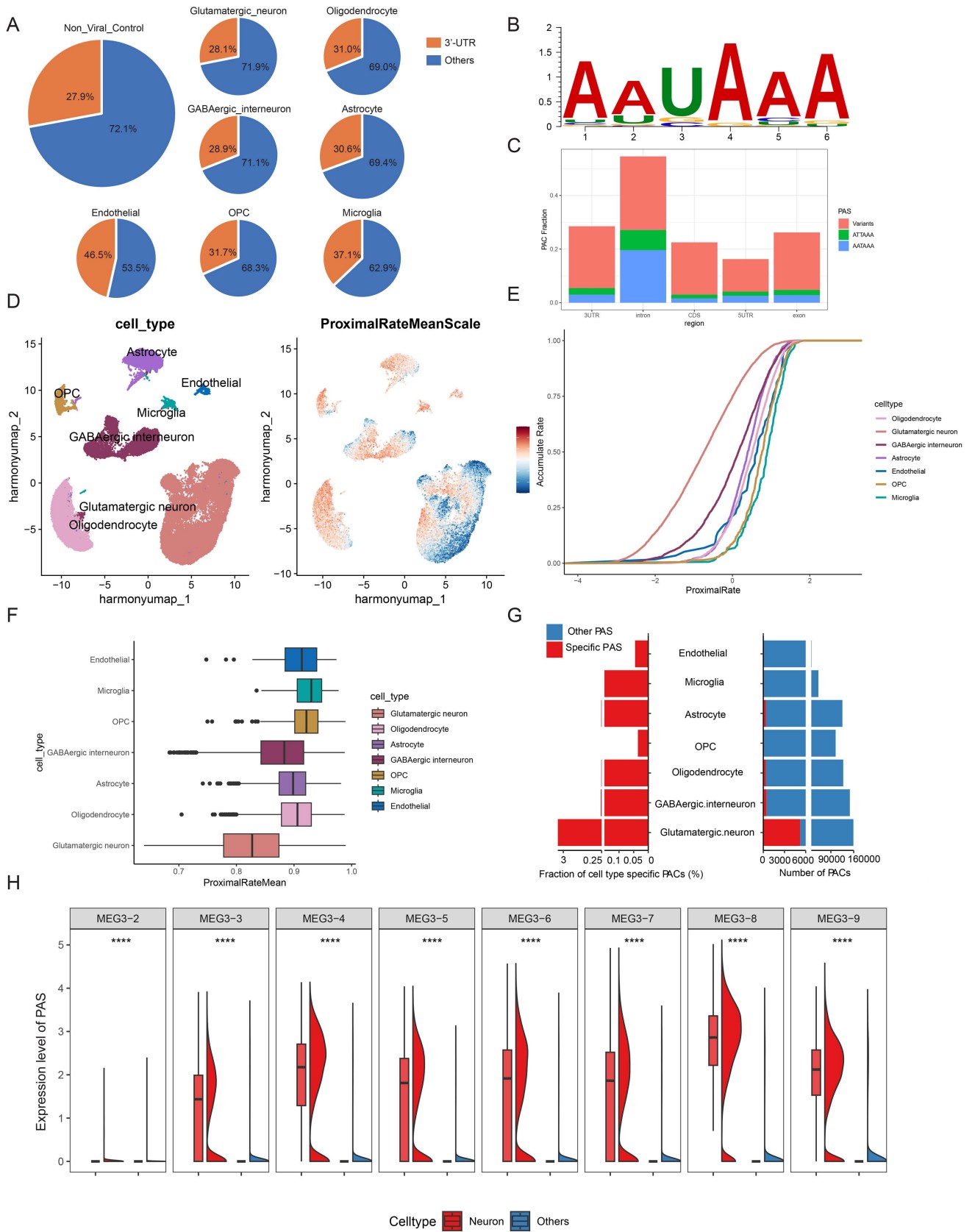

**Fig 1. Global APA levels in neural cells.** (A) Distribution of PAS in 3' UTR. (B) Canonical poly(A) motif (AAUAAA) enrichments for PAS. (C) Genomic distribution of PAS associated with the AAUAAA, AUUAAA, and single nucleotide variants of AAUAAA poly(A) signal categories within the upstream 50 nt of the PAS. (D) Left: UMAP of 16,020 nuclei from the medial frontal cortex of 7 control individuals, identified 7 different clusters. Each point depicting a single cell nucleus, colored according to cluster designation. Right: The mean PAS usage of each single cell for 7 samples after scaling, the expression level is from −3 to 3. (E) Accumulation rate of each cell type based on non-viral control group in solid lines. (F) Proximal rate of each cell type. Statistical comparison between neurons and other cell types was performed using the Wilcoxon rank-sum test (all p-values < 0.001; detailed values are provided in Supplementary Table 10.1 in S10 Table). (G) The numbers of cell-type-specific PAS and their fractions among the total identified PAS in different cell-type populations. PAS diversity values were calculated from normalized mean expression per cell type. (H) Expression levels of MEG3 PAS in neuronal cell types relative to other cell types. Boxplots were generated using standard statistical parameters: the boxes represent the interquartile range (IQR, 25th–75th percentile), the horizontal line within each box indicates the median, and the whiskers extend to 1.5 times the IQR from the lower and upper quartiles.

and polyadenylation [18]. The motif enrichment unraveled that the canonical poly(A) signal (AAUAAA) is significantly enriched in the dataset (Fig 1B). Similar to previous studies [3,19], AAUAAA, AUUAAA, and single nucleotide variants of AAUAAA were identified as typical poly(A) signals upstream of the PAS. Fig 1C depicts the genomic distribution of PAS associated with these three poly(A) signal categories, rather than the total PAS distribution shown in Fig 1A. These results support the authenticity of PAS identified by SCAPTURE (S1 Table).

We integrated seven samples of frontal cortex from individuals in the control group, generating 16,020 nuclei with 24,813 features. Unsupervised clustering identified seven major cell types (Fig 1D). The cell-type-specific markers uniquely expressed in each subpopulation were obtained from a previously published study [12]. Specifically, glutamatergic neurons are marked by SLC17A7, GABAergic interneurons are marked by GAD1, astrocytes are marked by GFAP, oligodendrocytes are marked by ST18, oligodendrocyte progenitor cell (OPC) is marked by VCAN, microglia are marked by CD74 and endothelial cells by CLDN5 (S1C Fig). S1A Fig shows that glutamatergic neurons represent the largest cell population (51%), followed by oligodendrocytes (18%) and GABAergic interneurons (14%). Astrocytes (9%) and OPC (5%) are present in moderate proportions, while endothelial cells (1%) and microglia (2%) are relatively sparse. Notably, two control samples (FC1 and FC6) exhibit a lower proportion of neurons and a higher proportion of oligodendrocytes compared with other individuals, which may reflect a combination of biological heterogeneity in sampling sites and technical variation. The top 10 most enriched genes in each cluster are shown in the heatmap of S1B Fig, and S1C Fig presents violin plots depicting the differential expression of these cell markers across cell types (S2 Table).

For glutamatergic neurons, the up-regulated gene sets are enriched in Alzheimer's disease (AD), and amyotrophic lateral sclerosis, belonging to calcium ion signaling and other related pathways, and conversely, down-regulated genes in astrocytes, OPC, oligodendrocytes, microglia, and endothelial cells are also enriched in these pathways. For GABAergic interneurons, genes are mainly enriched in metabolic pathways, the up-regulated genes are enriched in β-alanine, aspartate-alanine-glutamic acid and butanoate metabolism, and citrate cycle, and down-regulated in unsaturated fatty acid biosynthesis and adherens junction. For microglia, genes enriched in antigen processing and presenting are activated. These cells are more likely to participate in inflammation-related responses after infection (S1D Fig).

We calculated the APA usage in frontal cortex cells, the average PUI of each cell is projected into each cell type based on the gene expression in UMAP shown in Fig 1D. The pink color represents the scaled proximal rate mean of each cell, and the glutamatergic neurons and GABAergic interneurons have a relatively deep blue color referring to the mapping of cell subtypes, which indicates they prefer to choose the non-proximal PAS. These results are consistent with the previous reports that the neurons prefer to choose the distal PAS [2,20]. Subsequently, we described the accumulation plot and box plot of each cell type to further systematize the integrated APA usage, and we noticed that the average PUI of neurons is less than that of other cell types (Fig 1E, F). Furthermore, of all the PAS identified in various cell-type populations, neurons account for the greatest number as well as the largest fraction of cell-type-specific PAS (Fig 1G). Therefore, we focused more on changes in PAS levels in neurons compared to other neural cell types.

To address the functional consequences of APA events, we tested for the enrichment of gene ontology and functional pathway annotations among the genes with differential PAS used in neurons (excitatory neurons and inhibitory neurons) compared to non-neuronal nerve cells (S3 Table). Not surprisingly, these genes were highly enriched in biological processes relating to neural development or neural function (S2 Fig). In summary, we observed that a set of genes with neuronal regulatory functions was preferentially subject to distal PAS in neurons.

Neuronal cell types exhibited significantly higher expression of MEG3 PAS. MEG3 has been related to various processes such as necroptosis, apoptosis, inflammation, oxidative stress, endoplasmic reticulum stress, and epithelial-mesenchymal transition, according to recent studies [21–23]. In neurons, the usage of PAS sites was significantly higher in different MEG3 transcripts than in other neural cells (Fig 1H).

## Changes in global APA levels in neural cells following infection with SARS-COV-2

The control group (Fig 2A) and positive group (Fig 2C) display different patterns of preference, the shift to proximal in neurons is obvious after being infected with SARS-COV-2, and it indicates the wide changes in APA levels after infections. Each cell type proportion in the non-viral control group, COVID-19 group, and total are calculated. Fig 2B shows that excitatory neurons represent the most abundant cell type. Oligodendrocytes and inhibitory neurons are present in higher numbers compared to other cell types, while endothelial cells and microglia are relatively scarce. The accumulation plot of the infected group described that the neurons' average PUI is less than other cell types, but the difference becomes smaller after infections (Fig 1E, 2D).

Based on the above results, we further examined APA changes in each cell cluster and assessed statistical significance using the Kolmogorov-Smirnov test. In the cumulative curve plots (S3A, S3B Fig), glutamatergic and GABAergic neurons exhibited pronounced shifts toward proximal PAS usage in the COVID-19 group compared with the non-viral control group. In contrast, astrocytes (S3C Fig), oligodendrocytes (S3D Fig), OPCs (S3E Fig), microglia (S3F Fig), and endothelial cells (S3G Fig) showed only minor changes, with a slight tendency to shift toward non-proximal PAS usage after infection. This statistical analysis confirms that the differences observed in Figs 2A and 2C are supported by quantitative evidence rather than solely by visual inspection. For descriptive purposes only, the reported neurological symptoms of each COVID-19 donor are summarized alongside their sample-level APA metrics in Supplementary Table 10.6 in S10 Table.

The distinct trends observed after infection indicate that APA usage changes among different brain cell types. To further characterize these dynamics, we compared PAS events between the infected and control groups and visualized the results with a Sankey plot (Fig 2E), which illustrates the relationships among PAS genomic location, PAS type, and cell type. The left grey column represents the PAS location in the exon of genes (Fig 2E), and the 3'-UTR accounts for the largest proportion. The second column represents the PAS type. For analytical purposes, all non-proximal PAS were classified as distal PAS to facilitate calculation. The proportion of distal PAS usage is higher than that of proximal PAS, suggesting that brain cells preferentially utilize distal PAS. The third column is the cell type. We then summarized the number of significantly up- and down-regulated PAS in each cell type using bar plots (Fig 2F). Neurons, including glutamatergic neurons and GABAergic interneurons, showed a predominance of downregulated PAS, whereas astrocytes, oligodendrocytes, and OPC exhibited more upregulated PAS following infection (S4 Table). To account for the different number of testable PAS across cell types, normalized proportions and statistical comparisons are provided in Supplementary Table 10.2 in S10 Table, which confirmed the same trends.

The data in Fig 2E and 2F are consistent with the accumulation rate in the proximal rate shift in S3 Fig. In these two types of neurons, the increased proportion of downregulation PAS in proximal sites (Fig 2F) results in the smaller mean PUI after infections, which could explain why neurons have an increasing proximal accumulation rate after infection in S3A, S3B Fig. Besides, the other types of cells have a relatively low proportion of downregulated PAS in proximal sites and more upregulated PAS in distal sites, and the proximal accumulation rates are decreasing in these cell types (S3C-G Fig).

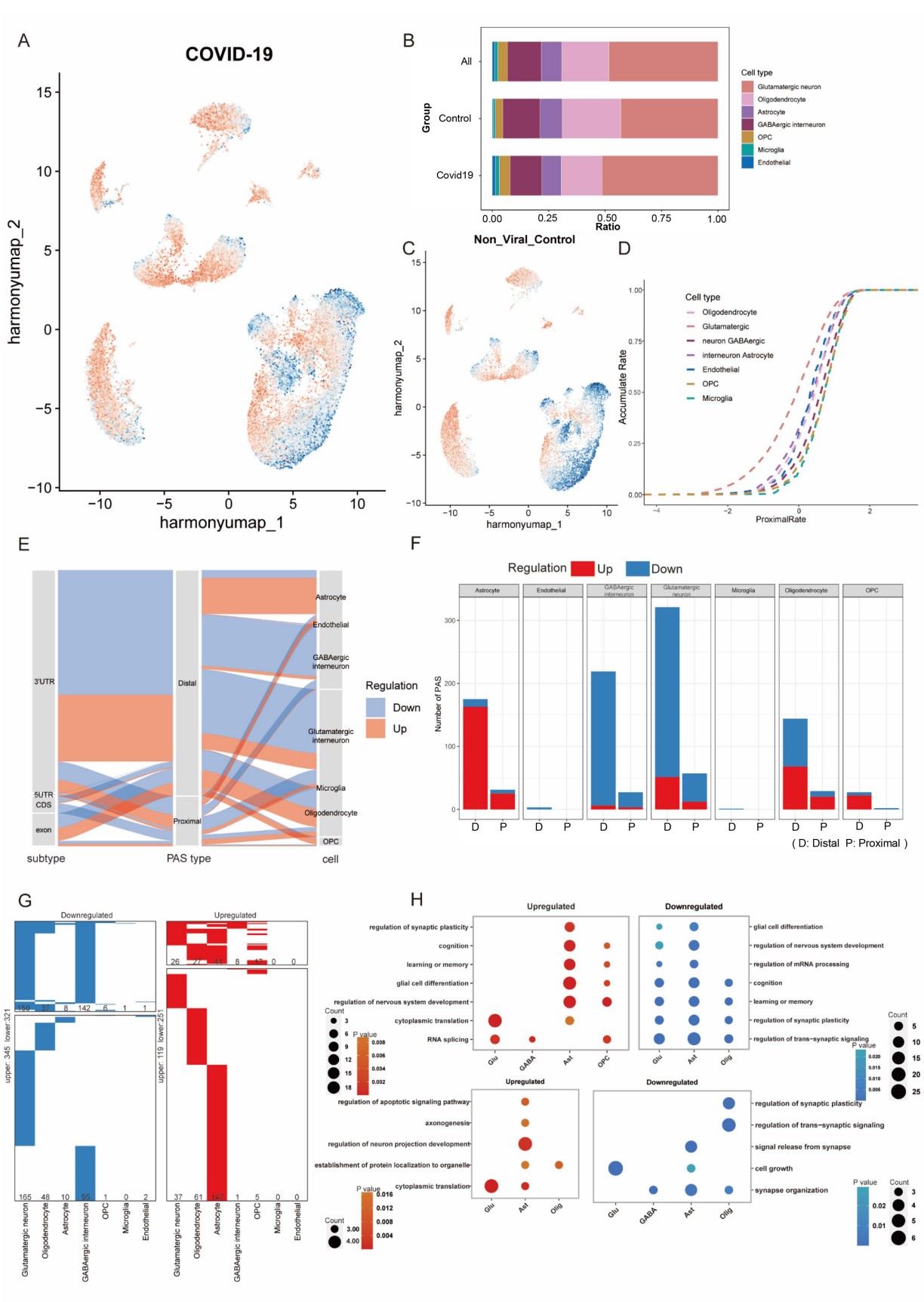

**Fig 2. Changes in global APA levels in neuronal cells following infection with SARS-COV-2.** (A) The global preference of APA usage in the infected group. (B) Each cell type proportion in the non-viral control group, COVID-19 group, and the combined dataset (All). (C) The global preference of APA usage in the non-viral control group. (D) Accumulation rate of each cell type based on the infected group with dashed lines. (E) Sankey plot showing relationships among PAS location, PAS type, and cell type. Pink indicates upregulated PAS and blue indicates downregulated PAS after infection. (F) Number of up- and down-regulated PAS per cell type after infection. Red bars indicate up-regulated PAS, and blue bars indicate down-regulated PAS. Statistical comparisons based on normalized proportions are provided in Supplementary Table 10.2 in S10 Table. (G) Heatmaps showing the distribution of upregulated (red) and downregulated (blue) PAS for each cell type in the brain between the non-viral control group and the infected group. PAS not differentially expressed are in white and the numbers of PAS are indicated. The upper part indicates the PAS shared by at least two cell types, and the lower panel indicates the unique PAS of each cell type. The numbers of PAS are annotated on the plots. (H) Dot plot showing the representative GO terms enriched for distal and proximal PAS in different cell types. The upper part indicates the distal PAS, and the lower panel indicates the proximal ones.

To validate the analysis results, we used the raw bam file from Cellranger, split it by cell types, and converted each file into the WIG format. The proximal and distal PAS usage was then inspected in IGV. Validation examples for selected PAS from each cell type, showing differential expression between the infected and control groups, are presented in S4 Fig. For each group, the first four samples are displayed, with peak heights in the sample tracks representing PAS expression levels. These visual results are compared with the SCAPTURE calculation.

The trends in PAS usage changes across these six cell types are consistent with the results shown in S5 Table. For instance, in oligodendrocytes, at the MT3 gene S4D Fig), the track heights in the control group are higher than those in the infected group, consistent with the results in S5 Table. Compared to the control group, the COVID-19 group exhibits reduced PAS usage in the MT3 gene, with an average log2 fold change of −1.1932.

To further investigate the effects of this significant PAS site usage at the cellular level, we compared the PAS expression of cell types between groups. To this end, we identified more than one thousand differentially expressed PAS ($|avg\_logFC| > 0.25$) in at least one cell type. 370 upregulated and 666 downregulated PAS were identified. Notably, approximately half of the differentially expressed PAS were specifically detectable in the neurons and astrocytes (Fig 2G).

Gene ontology (GO) analysis of differentially expressed PAS further uncovered that downregulated distal PAS in glutamatergic neurons was enriched largely in neuronal development and function (Fig 2H), such as synaptic plasticity, nervous system development, learning or memory, cognition, and so on, highlighting the importance of downregulated expressed distal PAS in neurons after infection, while downregulated expressed proximal PAS in astrocyte and oligodendrocyte were mainly associated with synaptic function after infection.

## The APA usage and its correlation with gene expression level

It has been reported that alternative PAS selection within the 3'-UTR can modulate post-transcriptional regulation and influence gene expression levels. We examined this relationship in glutamatergic neurons and GABAergic interneurons by integrating APA usage changes with differential gene expression results. We calculated APA usage by subtracting the usage value in the control group from the infected group using movAPA. We considered changes with an absolute value greater than 0.1 to be meaningful, following thresholds applied in previous studies [24] and taking into account the smaller effect sizes typically observed in single-nucleus datasets. The joint analysis enabled classification of genes into four categories corresponding to the quadrants in Figs 3A and 3B. In the first quadrant, APA usage value of the infected group is relatively larger, and the corresponding gene expression is increased; In the second quadrant, the APA usage value of the infected group is relatively small, and gene expression is increased; In the third quadrant, the APA usage value of the infected group was relatively small, and the gene expression was decreased; while in the fourth quadrant, both APA usage and gene expression are decreased.

In glutamatergic neurons (Fig 3A), the genes DDX17, RBM39, FNBP4, and others are located in the first quadrant indicating these genes prefer to use the distal PAS sites and the gene expression level increased. Genes like RPL37,

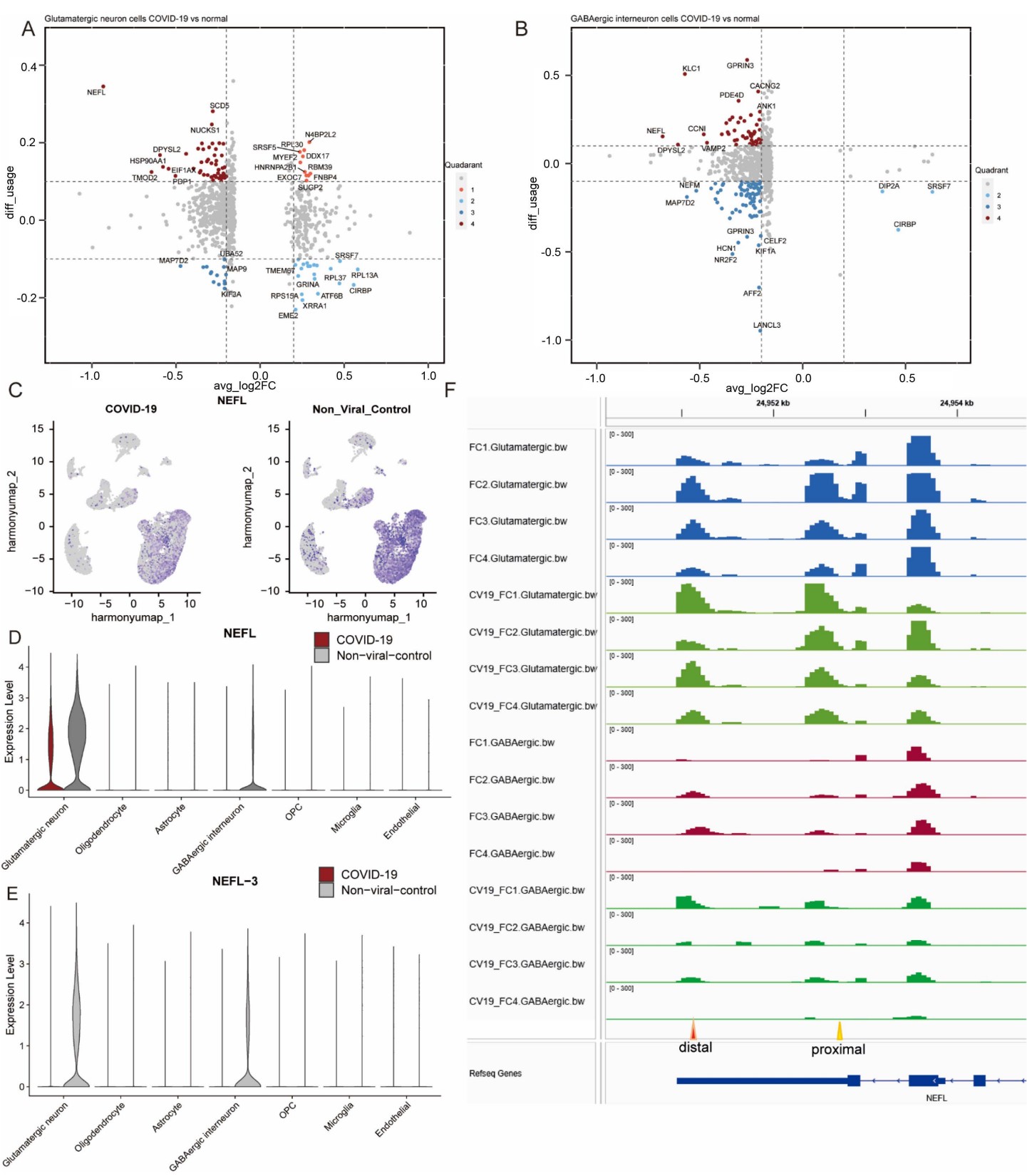

**Fig 3. The APA usage and its correlation with gene expression level.** Relationship between APA usage change (y-axis, infected – control, |Δusage|>0.1 considered significant) and gene expression change (x-axis, avg_logFC) in (A) glutamatergic neurons and (B) GABAergic interneurons. (C) Global gene expression of NEFL in the infected (left) and control (right) group. (D) NEFL gene expression in cell clusters. (E) NEFL-3 APA levels in cell clusters. The y-axis represents scaled PUI values. (F) The PAS usage of NEFL at distal and proximal sites by IGV.

RPL13A, GRINA, ATFB, and CIRBP located in the second quadrant prefer to use the proximal sites with increased gene expression. As for MAP7D2, MAP9 or others in the third quadrant tend to use the proximal sites, and gene expression is relatively downregulated in the infected group. The genes located in the fourth quadrant, such as NEFL, SCD5, and HSP90AA1, predominantly utilize distal PAS sites and exhibit downregulated gene expression level. For GABAergic interneurons in Fig 3B, we find that some genes have similar location patterns. For example, CIRBP is located in the second quadrant in these two types of cells, and NEFL in the fourth quadrant, for both cell types. For other cell types, NEFL is also predominantly located in the fourth quadrant (S5 Fig, S6 Table).

NEFL is a light chain of neurofilaments that functionally maintain the neuronal caliber. Individuals with dementia have elevated levels of NEFL in the cerebrospinal fluid (CSF) [25,26]. In Fig 3C, UMAP visualization is used to illustrate the spatial distribution of NEFL expression across cell types, showing that NEFL is mainly enriched in glutamatergic neurons and GABAergic interneurons, with a visible reduction in the COVID-19 group. The quantitative gene expression levels for each cell cluster are provided in Fig 3D, which confirms a significant decrease in NEFL expression in the COVID-19 group compared to controls. For APA usage (Fig 3E), the PAS site usage in NEFL-3 is decreased after infection. In our APA analysis, NEFL-3 refers to the third annotated PAS within the NEFL transcript. The PAS usage results (Fig 3E) show that NEFL-3 usage is decreased after infection, suggesting that COVID-19 may alter NEFL transcript processing at the 3'-UTR level, potentially contributing to the observed expression changes.

To validate the result of NEFL, we visualized the abundance of PAS sites using wig file of excitatory neurons and inhibitory neurons obtained from raw bam file by IGV shown in Fig 3F.

NEFL is located on the negative strand of the chromosome, with the 5'-UTR positioned on the right and the 3'-UTR on the left. Proximal PAS sites are marked with orange triangles, and distal PAS sites with red triangles in Fig 3F. After infection, the distal PAS sites show higher average peak heights relative to total PAS usage compared with the control group, particularly in sample CV19-FC3. This observation is consistent with the trend shown in Figs 3A and 3B, indicating a shift toward distal PAS usage in NEFL following infection.

### Genes with variational APA levels and differential expression widely affected neural function in microRNA-dependent fashion

Previous studies have demonstrated that APA events can influence gene regulation in a microRNA (miRNA)-dependent fashion. Based on this knowledge, we assessed the potential impact of APA events on miRNA binding by identifying miRNA target sites gained or lost in the 3'UTR after APA changes in infection samples [27,28]. In infected samples, APA results in the inclusion or exclusion of miRNA target sites within gene 3'-UTRs.

To determine which miRNAs might be affected, we applied three curated databases-miRcords [29], miRTarBase [30], and TarBase [31]-to the genes showing APA changes and identified 2,404 miRNAs whose target sites are predicted to be impacted (S7 Table) in the four quadrants across all cell types. Figs 4A and 4B summarize these predicted changes, showing the numbers of miRNA target sites gained or lost in each major cell type as a result of APA shifts. Nearly half of these miRNAs being cell-type-specific (Fig 4C, 4D).

In order to further investigate the effects of these genes, we examined their distribution across cell types. In quadrant 2, the largest numbers of genes were observed in astrocyte and oligodendrocyte (31 and 51 genes), while glutamatergic and GABAergic interneurons contained 50 and 51 genes, respectively, in quadrant 4 (Fig 4E). Functional annotation

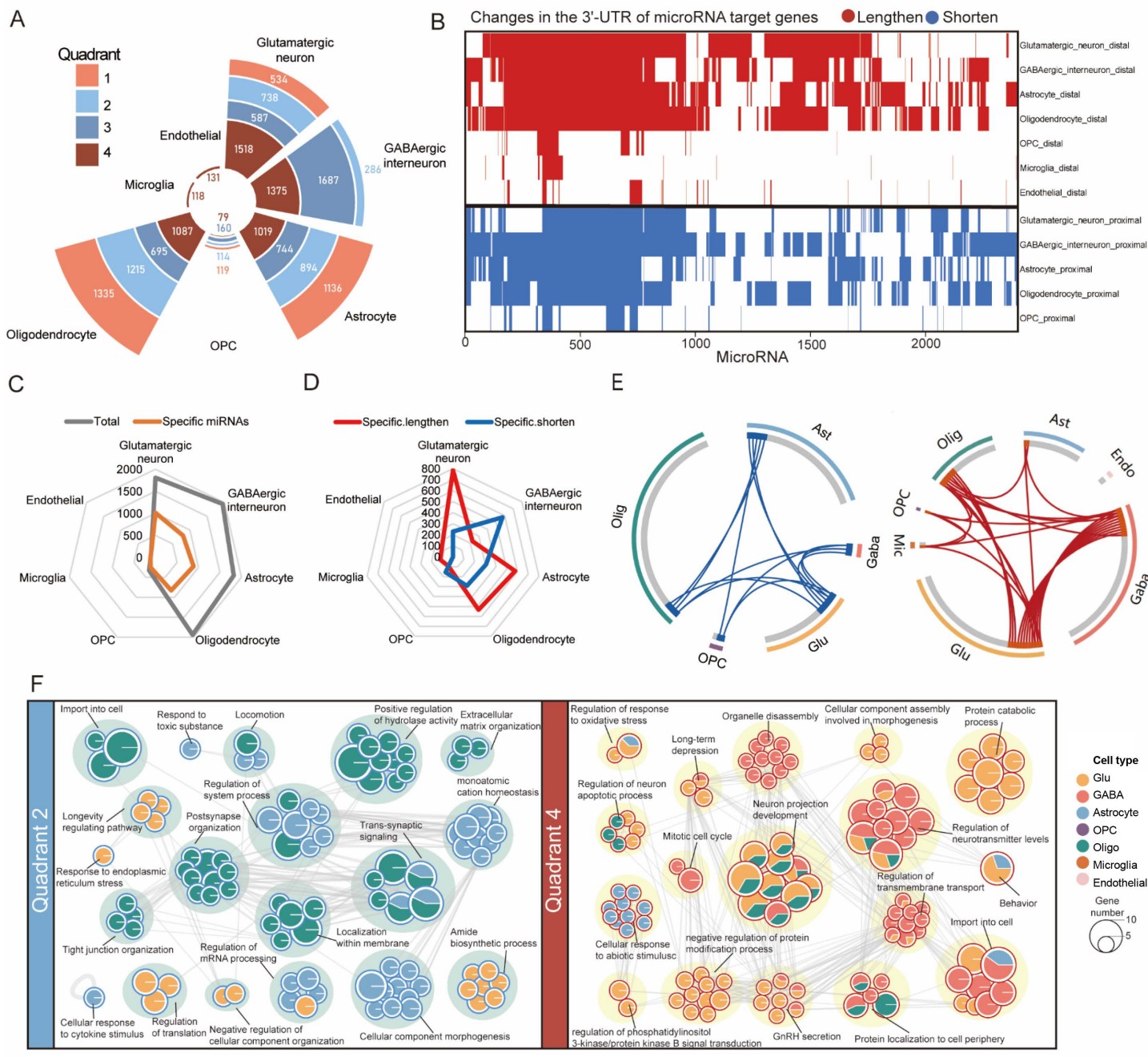

**Fig 4. Genes with variational APA levels and differential expression widely affected neural function in a microRNA-dependent fashion.** (A) Radial plot showing the number of microRNAs targeting the four quadrants of genes. (B) Heatmap showing the changes in the 3'-UTR of microRNA target genes, either lengthen (red) or shorten (blue). (C) Radar Charts showing the total, (D) specific microRNAs targeting lengthen or shorten 3'-UTR across cell types. (E) Circos plots show the overlap of genes from different cell types in quadrant 2 (left) and quadrant 4 (right). Connecting lines link the same genes that are shared by cell types. (F) GO terms and pathways of quadrant 2 (left) and quadrant 4 (right) genes in different cell types. The network nodes were displayed as pies. Each pie sector is proportional to the number of hits originating from a gene list. The pie charts are colored by cell types, where the size of a slice represents the percentage of genes under the term that originated from the corresponding cell type. Terms with a similarity score > 0.3 are linked by an edge (the thickness of the edge represents the similarity score).

enrichment analysis of these predicted target genes revealed terms related to neuronal processes, including "neuron projection development" and "regulation of neurotransmitter levels" in quadrant 4 for glutamatergic neurons, GABAergic interneurons, and oligodendrocyte, "long-term depression" and "GnRH secretion" in both glutamatergic neurons and GAB-Aergic interneurons, and "trans-synaptic signaling" and "regulation of system process" in astrocyte and oligodendrocyte in quadrant 2. "RNA splicing", "glial cell differentiation", and "Parkinson disease" were enriched in quadrant 1 (Fig 4F, S6 Fig, S8 Table). These findings are predictive in nature, and further biological assays will be required to confirm the APA–miRNA regulatory mechanisms.

### Neurological and psychiatric disease risk genes are significantly altered in APA following infection

To unravel the potential mechanisms underlying the development of neurological and psychiatric effects following SARS-COV-2 infection, fifteen common neurological and psychiatric disorders, including AD, Parkinson's disease (PD), epilepsy, autism spectrum disorders (ASD), depressive disorder, bipolar disorder (BD), attention deficit hyperactivity disorder (ADHD), multiple sclerosis (MS), ischemic stroke, schizophrenia (SZP), amyotrophic lateral sclerosis (ALS), Huntington disease (HD), migraine disorder, brain aging, and anxiety, were selected for joint analysis using the two databases DisGeNET [32], BioKA [33] and diseases from the GWAS [34] catalogue in the published paper [12]. After compiling all genes in the four quadrants showing variations in APA levels and differential expression across the seven cell types (glutamatergic neuron, GABAergic interneuron, astrocyte, endothelial, microglia, oligodendrocyte, and OPC), we identified a total of 408 genes, of which 267 were classified as risk genes according to the above databases (S9 Table; Fig 5A, 5B). These 267 genes showed significant enrichment for neurological disorders and traits, particularly AD, PD, and schizophrenia. For example, CALM1, a component of the calcium signal transduction pathway [35], was downregulated accompanied with a shorter 3'-UTR after SARS-COV-2 infection (S7A, S7B Fig). A recent study demonstrated that elimination of the *Calm1* long 3'-UTR mRNA isoform via CRISPR-Cas9 impairs dorsal root ganglion development and hippocampal neuron activation in mice [36]. The amyloid precursor protein (APP) has received considerable attention due to its proposed role in the pathogenesis of AD [37,38]. We discovered that expression of APP increased after infection, with oligodendrocytes displaying a longer 3'-UTR and astrocytes a shorter 3'-UTR (S7C, S7D Fig). It has been verified that APP 3'UTR length can affect translation efficiencies. A variation in the length of the APP 3'UTR, whether long or short, can cause aberrant expression and eventually increase the risk of AD [39]. Actin constitutes a prominent cytoskeletal protein within eukaryotes organisms, and is recognized for its localization in neuronal synapses. Investigation into beta-actin (ACTB) reveals that two alternative transcripts were induced and terminated at tandem polyA sites, the longer 3'UTR brings higher stability to the transcript, resulting in higher gene expression, in mouse neuronal cells [40]. In our study, the expression of ACTB increased after infection, with a longer 3'-UTR in astrocytes (S7E, S7F Fig). TMOD2 acts as a fibrogenic gene in idiopathic pulmonary fibrosis (IPF) and exhibits a shortened 3'UTR and increased protein expression levels [41]. In the present study, after SARS-COV-2 infection, TMOD2 shows 3'-UTR shortening as well as expression downregulated.

The remaining 141 genes that were not assigned to known disease-associated gene sets are listed in Supplementary Table 10.3 in S10 Table, and their GO functional enrichment analysis is provided in Supplementary Table 10.4 in S10 Table. Although these genes are not linked to known brain disease risk in current databases, they are involved in essential molecular and cellular functions-such as regulation of RNA processing, viral processes, cilium assembly, and translation-that could be relevant to cellular responses and pathophysiological processes during infection.

## Discussions

There is growing evidence that SARS-CoV-2 infection causes neurological deficits in a large proportion of infected patients [6,42,43]. Several studies have reported psychiatric symptoms in patients with COVID-19, and a growing body of research suggests that brain disorders may persist after recovery from primary infection [44,45]. Though it has been reported that the gene expression level changes after infections [46,47], the changes in APA level have not been reported

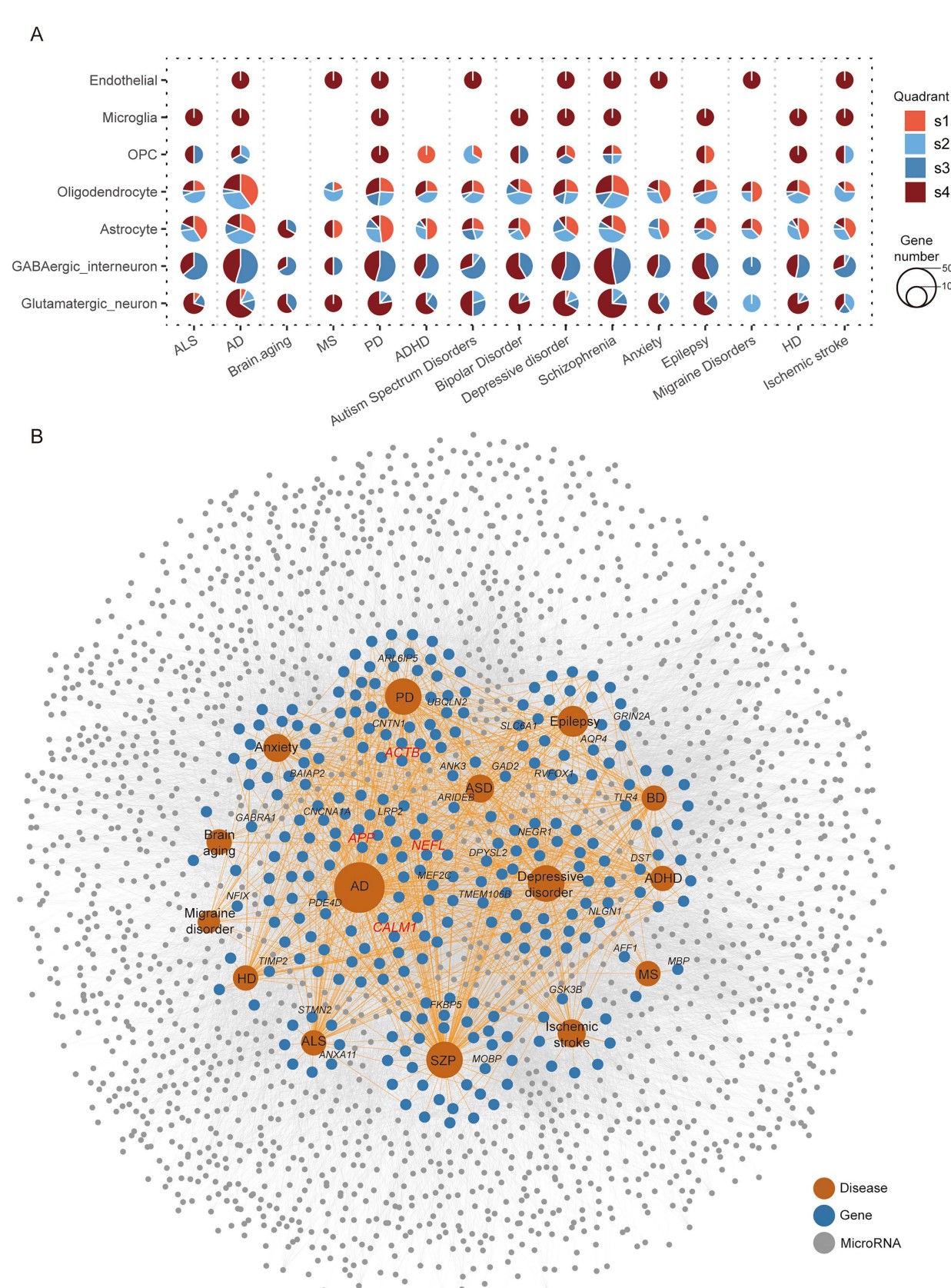

**Fig 5. Neurological and psychiatric disease risk genes are significantly altered in APA following infection.** (A) Dot plots showing that genes from four quadrants overlapped with genes from neurological and psychiatric disorders associated gene sets. The pie charts are colored by quadrants, with size indicating the frequency of genes. (B) Network visualizing the overlap between the genes in the four quadrants and genes involved in 15 common neurological and psychiatric disorders. The colors of grey, blue, and orange nodes represent targeting miRNAs, genes and diseases, respectively. The colors of grey and light orange lines represent the connection between genes-miRNAs and genes-diseases.

in the brain tissue after infection. We selected an appropriate analytical approach and identified significant changes in global APA levels (Fig 2).

With shortened 3'-UTR average length and extensive APA response to viral infection, expression levels of genes with APA are altered and enriched in immune-related pathways, such as Toll-like receptor, RIG-I-like receptor, JAK-STAT, and apoptosis-related signaling pathways [11]. In a genome-wide analysis with PBMC RNA-seq dataset of COVID-19 patients [48], APA-related genes are abundant in the ontology classification related to innate immunity, such as neutrophil activation, MAPK cascade regulation and cytokine production, and interferon-γ and innate immune response, suggesting that APA events may be a better predictor of neurological deficits than alternative splicing for COVID-19 patients. Therefore, APA could be considered a potential therapeutic target and a novel biomarker for post-COVID survivors.

In our study, we find that the average utilization of proximal PAS in glutamatergic neurons and GABAergic interneurons increases after COVID-19 infection, indicating that brain cells have a broad APA response after SARS-CoV-2 infection, consistent with APA changes observed during other virus infection [48,49]. The selection of PAS varies among cell subtypes. In neurons, the control group primarily uses distal PAS, in agreement with previous reports [20]. In contrast, both microglia and endothelial cells predominantly express transcripts with short 3'-UTRs [20]. For COVID-19, the excitatory neurons and inhibitory neurons show a relative shift toward distal PAS. Importantly, our observations do not imply neuronal cell death. As reported by Sommerkamp et al.[50], shorter 3'UTRs and increased proximal PAS usage can also be a feature of neural stem/progenitor cells or proliferative states, whereas differentiated neurons typically exhibit longer 3'UTRs. Therefore, the proximal shifts observed here may reflect changes in molecular identity or stress-adaptive programs rather than loss of neurons.

Although our study does not experimentally dissect causal pathways, prior work has shown that viral infections can modulate APA through multiple molecular processes, including changes in the expression or activity of core 3'-end processing factors, RNA-binding proteins, and transcriptional elongation dynamics [11,51,52]. For example, innate immune activation can alter the CPSF [51], thereby favoring proximal PAS usage. Moreover, coronaviruses, including SARS-CoV-2, can directly or indirectly influence host transcription termination and RNA stability [53], potentially contributing to the APA landscape changes we observe.

The genes with different APA levels deserve further study, especially in glutamatergic neurons and GABAergic neurons. We extract all PAS in different cell types for further GO analysis to investigate related gene expression changes following infection, and the impacts of APA levels on altering cellular pathway. We find that APA changes are related to RNA splicing, and neuronal development and function (Fig 2H), such as synaptic plasticity, nervous system development, learning or memory, and cognition. GO analysis suggests that brain function may be impaired due to global gene expression and APA changes, which could explain the neuropsychiatric symptoms and long-term brain sequela observed in some infected individuals [54].

The results of APA usage and its correlation with gene expression (Fig 3) could be divided into four quadrants. About 90% of genes in the long isoform are less stable and produce less abundant proteins, but this is not universal because because APA can have different regulatory effects on mRNA stability and translation [55]. Genes located in the second quadrant (Fig 3) are related to the short isoform, which is more stable and has higher expression, while those in the fourth quadrant represent the long isoform and are less stable.

As for APP, variations in the length of the APP 3'UTR, whether long or short, can influence its expression in a context-dependent manner, potentially contributing to AD risk. For example, Mbella EG et al.[39] found that in their experimental system, alternative polyadenylation of APP mRNA generates isoforms with different 3′UTR lengths and translation efficiencies, with the long 3′UTR enhancing translation compared to the short form. However, such effects are not universal, as 3′UTR length can either enhance or inhibit translation depending on the presence of specific RNA-binding proteins, microRNA target sites, and RNA secondary structures. Certain proteins from the human brain, CHO cells, and Xenopus oocytes bind only to the short 3'UTR but not the long one [39]. These findings suggest a relationship between the effectiveness of mRNA translation and protein binding to the 3'UTR of APP mRNA. The existence of secondary structures in the long nucleotide sequence may be the cause of the absence of these particular interactions between the protein and the long 3'UTR. Either long or short, deviations in APP 3'UTR length can result in aberrant expression and eventually lead to increased disease risk.

A total of 267 risk genes for common neurological and psychiatric disorders were found to undergo significant changes in APA following infection, implying that APA could be a potential therapeutic target and predictive biomarker for COVID-19-associated neurological and psychiatric sequela.

Crosstalk between miRNAs and APA is involved in numerous aspects of gene versatility and cell cycle [56]. For instance, the gene cell division cycle 6 (CDC6) is essential for DNA replication. In mammalian cells, CDC6 can govern the onset of DNA replication and limit the rate of S-phase entry [57]. Estrogen can cause the 3'UTR of CDC6 to become shorter, producing isoforms that evade miRNA-mediated repression, leading to aberrant CDC6 expression. Consequently, investigating potential miRNA regulation in the context of APA is promising. A total of 2404 miRNAs targeting the 408 genes in the four quadrants (Fig 4A, 4B) were predicted using the miRcords, miRTarBase, and TarBase. The miRNAs' target site information can indicate the impact of APA events and has the potential to be used as a predictive biomarker.

In conclusion, after the SARS-CoV-2 infection, APA usage in neural cells in brain tissue changes in a cell-type-specific manner. These changes are enriched in genes related to RNA splicing, translation, synaptic function, and nervous system function, and are associated with neurodegenerative and mental diseases.

Nevertheless, our study has certain limitations. First, our analyses were based on brain samples from SARS-CoV-2–infected and control individuals, without including additional disease controls such as non-SARS-CoV-2-infected patients with other illnesses, or longitudinal samples from the same patients after recovery. These additional cohorts would help distinguish APA changes that are specifically attributable to viral infection from those that may reflect general sickness, inflammation, or treatment effects. Unfortunately, such samples were not available for the present study. In the future, we plan to extend our work by incorporating both disease-control and post-recovery cohorts when appropriate datasets become available, which will enable a more precise dissection of virus-specific versus non-specific APA regulation.

## Supporting information

**S1 Fig. The cell identity analysis by gene expression.** (A) Each cell type proportion in 7 control individuals. (B) Heatmap plot showing the top 10 most differentially upregulated genes in each cell type identified through unsupervised clustering. Red indicates higher expression; Grey indicates lower expression; the Average expression (avg. exp) scale is shown on the top. (C) Violin plot showing the expression level of cell markers with significantly different percentages per cell type. (D) Heatmap plot showing co-upregulated or co-downregulated gene sets per cluster in RRA. Red indicates up-regulated; Blue indicates down-regulated.
(TIF)

**S2 Fig. Distal PAS contributions to neural function.** GO terms and pathways enriched among genes with differential PAS usage in neurons compared to non-neuronal cells. The size of each network node represents the number of genes associated with the corresponding term. Terms with a similarity score＞0.3 are connected by edges.
(TIF)

**S3 Fig. The Global preference of proximal PAS usage in cell types based on the accumulation rates.** Scaled proximal usage index (PUI) values (x-axis, ranging from –3–3) are plotted for different cell types, with values representing normalized expression levels. The y-axis shows the accumulation rate. Grey lines indicate the accumulation rate for each cell type in the non-viral control group, and red lines indicate the COVID-19 group. Statistical significance was assessed by Kolmogorov-Smirnov test, and p values are indicated for each cell type: (A) glutamatergic neurons ($p < 2.2e-16$). (B) GABAergic interneurons ($p < 2.2e-16$). (C) oligodendrocyte ($p = 1.282e-9$). (D) OPC ($p = 1.623e-2$). (E) astrocytes ($p = 1.238e-8$). and (F) microglia ($p = 1.293e-3$).
(TIF)

**S4 Fig. The PAS validation by IGV in each cell type.** (A) RPL37A gene in glutamatergic neurons. (B) CIRBP gene in GABAergic interneurons. (C) F3 gene in astrocytes. (D) MT3 gene in oligodendrocytes. (E) B2M gene in OPC. (F) JUND gene in microglial. The same color means that the samples belong to the same group; all the tracks keep the same data range, and the triangle at the bottom indicates the PAS sites.
(TIF)

**S5 Fig. APA usage and its correlation with gene expression level of (A) Astrocyte and (B) Oligodendrocyte.**
(TIF)

**S6 Fig. GO terms and pathways of quadrant 1 (upper) and quadrant 3 (lower) genes in different cell types.** (A) The network nodes were displayed as pies. Each pie sector is proportional to the number of hits originating from a gene list. The pie charts are colored by cell types, where the size of a slice represents the percentage of genes under the term that originated from the corresponding cell type. Terms with a similarity score > 0.3 are linked by an edge (the thickness of the edge represents the similarity score).
(TIF)

**S7 Fig. Network visualizing the connection between the (A) _CALM1_, (C) _APP_, (E) _ACTB,_ and common neurological and psychiatric disorders.** The colors of grey, blue, and orange nodes represent targeting miRNAs, genes, and diseases, respectively. Relative gene expression and APA levels of (B) _CALM1_ in glutamatergic neurons, (D) _APP_ in astrocytes, and (F) _ACTB_ in astrocytes, comparing the control group to the COVID-19 group.
(TIF)

**S1 Table. PAS metadata.**
(XLSM)

**S2 Table. The top 20 markers of clusters.**
(XLSX)

**S3 Table. PAS changes in neuronal cell types relative to other cell types in control group.**
(XLSX)

**S4 Table. PAS changes in different cell types after infection.**
(XLSX)

**S5 Table. PAS changes in different cell types from SCAPTURE results.**
(XLSX)

**S6 Table. The 408 genes in the four quadrants of all cell types.**
(XLSX)

**S7 Table. The predicted microRNA targeting the 408 genes in the four quadrants of all cell types.**
(XLSX)

**S8 Table. GO terms and pathways of the genes in the four quadrants of all cell types.**
(XLSX)

**S9 Table. The 267 identified risk genes in DisGeNET and BioKA databases and GWAS catalogue.**
(XLSX)

**S10 Table. This file contains Supplementary tables 10.1–10.7, including: 10.1.** Wilcoxon test p-values comparing ProximalRateMean between neurons and each non-neuronal cell type in the control group. **10.2.** Proportion of significant PAS across cell types and statistical comparisons. **10.3.** The 141 genes retrieved from 4 quadrants but are not assigned to disease risk genes. **10.4.** GO functional enrichment analysis of the 141 genes. **10.5.** Summary of key donor metadata. **10.6.** Summary of neurological symptoms and average PUI per COVID-19 sample overall and by major cell type. **10.7.** Summary of sample-level QC metrics.
(XLSX)

## Acknowledgments

The authors would like to thank professor Baofa Sun for his support during the research process.

## Author contributions

**Conceptualization:** Wanshan Ning, Jingjing Yang.

**Data curation:** Qun Chen, Shuai Liu.

**Formal analysis:** Ying Gu.

**Investigation:** Qun Chen, Xingyu Li, Ruizhi Xu, Ruixi Ye, Jingjing Yang.

**Methodology:** Wanshan Ning, Qun Chen, Ying Gu.

**Project administration:** Wanshan Ning, Jingjing Yang.

**Resources:** Ruizhi Xu.

**Software:** Qun Chen.

**Supervision:** Wanshan Ning, Xingyu Li.

**Validation:** Shuai Liu.

**Visualization:** Qun Chen, Ying Gu, Shuai Liu, Ruixi Ye.

**Writing – original draft:** Qun Chen, Ying Gu, Shuai Liu.

**Writing – review & editing:** Wanshan Ning, Jingjing Yang.

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
