## [Decision Letter · Decision Letter 0]

7 Jul 2025

Dear Dr. Ning,

Thank you for submitting your manuscript to PLOS ONE. After careful consideration, we feel that it has merit but does not fully meet PLOS ONE’s publication criteria as it currently stands. Therefore, we invite you to submit a revised version of the manuscript that addresses the points raised during the review process.

We look forward to receiving your revised manuscript.

Kind regards,

Milad Khorasani, PhD

Academic Editor

PLOS ONE

Journal Requirements:

6. We are unable to open your Supporting Information file “Supplementary table 2-9.xlsm”. Please kindly revise as necessary and re-upload.

Reviewers' comments:

Reviewer's Responses to Questions

**Comments to the Author**

1. Is the manuscript technically sound, and do the data support the conclusions?

Reviewer #1: Partly

Reviewer #2: Partly

Reviewer #3: Partly

Reviewer #4: Yes

2. Has the statistical analysis been performed appropriately and rigorously?

Reviewer #1: Yes

Reviewer #2: Yes

Reviewer #3: No

Reviewer #4: Yes

3. Have the authors made all data underlying the findings in their manuscript fully available?

Reviewer #1: Yes

Reviewer #2: Yes

Reviewer #3: Yes

Reviewer #4: Yes

4. Is the manuscript presented in an intelligible fashion and written in standard English?

Reviewer #1: No

Reviewer #2: No

Reviewer #3: No

Reviewer #4: No

Reviewer #1: General Assessment

The authors present an interesting analysis of single-cell alternative polyadenylation (APA) in brain tissues from COVID-19 patients compared to controls. The manuscript explores APA events in different brain cell types and suggests possible mechanisms linking APA changes to neurological and psychiatric sequelae observed after SARS-CoV-2 infection. This is a timely and relevant topic. However, the manuscript requires significant improvements in writing clarity and more robust mechanistic interpretation to fully support the conclusions.

Major Comments

1. APA vs Neurological Symptoms Correlation

The manuscript would be strengthened by explicitly correlating the observed APA/PAS changes in each COVID-19 sample with the neurological symptoms of each corresponding patient. For instance, the supplementary data from Yang et al. (2021, Wyss-Coray Lab) includes patient-level neurological assessments that could be referenced.

2. Lack of Evidence for Neuronal Cell Death

The manuscript implies functional impairment in neurons after infection, but does not provide direct evidence of neuronal death. Are neurons dying, or merely losing their molecular identity by downregulating distal PAS usage typical of differentiated neurons? This distinction should be clarified with appropriate data or cautious interpretation. Proximal PAS and shorter 3’UTRs are present in stem cells and dividing cells, for example. (Reference: Sommerkamp P, Cabezas-Wallscheid N, Trumpp A. Alternative Polyadenylation in Stem Cell Self-Renewal and Differentiation. Trends Mol Med. 2021 Jul;27(7):660-672. doi: 10.1016/j.molmed.2021.04.006. Epub 2021 May 11. PMID: 33985920.)

3. Mechanistic Insight Lacking

The study does not clearly elucidate the mechanism by which SARS-CoV-2 infection leads to the observed changes in APA and PAS site usage. This weakens the causal link between viral infection and APA dysregulation. Further experimental validation (or at least deeper discussion of known molecular mechanisms) would be helpful.

4. Focus on Neurons vs Astrocytes

The authors report that astrocytes exhibit more unique upregulated PAS events than neurons. Why then is the primary focus placed on neurons? A rationale for this choice should be discussed.

5. Interpretation of NEFL Data

The NEFL gene is highlighted as showing a switch to distal PAS, yet this is clearly demonstrated in only one of four samples. The evidence should be presented more cautiously, acknowledging variability.

6. APP 3’UTR Interpretation

In the Discussion (line 532), the claim that longer 3’UTRs increase APP translation needs more nuance. In wild-type contexts, longer 3’UTRs can enhance or inhibit translation depending on context. The problem in AD arises with mutant APP aggregation, not merely expression changes of the wild-type gene. Please clarify.

Minor Comments

• The writing throughout the manuscript contains multiple grammatical and syntactical issues that impact clarity. Specific problematic lines include (but are not limited to): 76, 80, 105–107, 187, 192, 219, 254, 271–273, 298, 300–301, 393, 451, 501, and 562. A thorough English language edit is necessary.

• The sentence in lines 341–343 should be revised for clarity. Suggested rewrite:

“We calculated APA usage by subtracting the usage value in the control group from the infected group. We considered changes with an absolute value greater than 0.1 to be significant.”

• A reference should be added for the movAPA software used in the analysis.

• Consider updating the running title to:

“Effects of COVID-19 on single-cell alternative polyadenylation”

for better clarity and relevance.

• The reviewer was unable to access some supplementary tables due to possible malware alerts. Ensure all supplementary materials are securely hosted and accessible.

• Supplementary Figure 1: The proportion of neurons varies dramatically among the 7 control individuals. If these samples were taken from similar brain regions, such inter-individual variability should be minimal. Could the authors comment on potential batch effects or anatomical heterogeneity?

• Supplementary Figure 2: The figure lacks adequate explanation. Specifically, it is unclear what the circle sizes represent. Please clarify this in the figure legend and/or main text.

• Supplementary Figure 3: The y-axis label is missing or unclear. What parameter is being plotted (e.g., Proximal Usage Index, expression level, etc.)? This should be clearly defined.

• Supplementary Figure 7: It is unclear whether the data presented reflect all cell types combined or specific cell subtypes. Please indicate which cell populations are represented in the plots and whether differences exist across subtypes.

Editorial Criteria Responses

1. Is the manuscript technically sound, and do the data support the conclusions?

Partly. The bioinformatic analyses are sound, but some conclusions are overstated relative to the evidence provided.

2. Has the statistical analysis been performed appropriately and rigorously?

Yes.

• Have the authors made all data underlying the findings in their manuscript fully available?

Yes, but the files for supplementary tables have to be revised due to malware alerts.

3. Is the manuscript presented in an intelligible fashion and written in standard English?

No. The manuscript requires substantial editing for grammar and clarity.

Recommendation:

Minor Revision

While the manuscript presents valuable data, improvements in writing, clarification of biological interpretations, and cautious framing of key claims are needed to meet PLOS ONE's publication standards.

Reviewer #2: In this paper, the authors used a pre-published dataset to investigate the impact of a SARS-CoV-2 infection on the Alternative Polyadenylation patterns in the brain.

They start with monitoring the global alternative polyadenylation landscape in the neural cells. Then, they address the changes in this landscape in the context of an infection. It allows them to correlate the APA usage to gene expression, next correlated to predicted miRNA binding. Finally, they propose to use APA as biomarkers of future clinical consequences of SARS-CoV-2 infections.

General comment :

The study makes sense and might raise interest in the specific community of researchers working on APA in infectious context related to neurological disease. It could be published after revision.

The English level is appropriate, even though some sentences (reported in the subsequent sections) should be corrected before publication. The most notable example is the consistent use of anthropomorphic phrases. "choose" and "prefer" are not appropriate when it comes to genes/transcripts/inanimate entities.

Revisions (marked with +, ++, +++ or ++++ to determine their importance):

++++ In a general manner, the authors tend to be obscure in the way they are choosing, interpreting or reporting results. Also, figure legends do not seem to be complete. Here are examples that should be clarified :

+ Line 35 : "some prefer short isoform" should be rephrased

+ Line 66-68 : Could the authors explain, or clarify? They first say the genome is more broadly expressed in neurone, explaining the large mRNA diversity and then; they say the large mRNA diversity is due to alternative posttranscriptional mechanisms...

+ Line 108 : rephrase in appropriate English

+ Line 113-114 : rephrase in appropriate English

++ Lines 140-146 : The authors explain the PUI concept. While the metric might be appropriate, the authors do not explain appropriately how it is relevant as a metric. They also should clarify if they only used genes including only two PAS since it is what is implied by their phrasing.

+ Lines 149-151 : The author should clarify their choice in the transcripts (p value or p adj? why log2 fold change larger than 0.25?)

+ Line 174 : "poly(A)" is a confusing term in that case because it refers to polyadenylation while it usually refers to polyadenosine stretches. It should be changed for clarity.

+++ Figure 1G - Lines 216-220 : The authors show that some cell types use more diverse PAS than others. However, the population showing the highest diversity are also the most abundant. This abundance increases the probability of detecting low abundance APA. The authors should use cell subsets of the same size to determine the APA abundance instead of using the entire cell subpopulations.

+ Lines 244-245 : The legend should be completed for the readers to know what are the values behind each box of the boxplot (interquartile range? 5%-95%? Min-Max?)

+ Line 252-255 : Can the authors comment on the decrease in the oligodendrocyte population from non-infected to infected?

+++ Line 261 : "relatively high significant preference"  no mention of the statistics are made anywhere + "preference" is not appropriate + the sentence is hard to read

++++ Lines 266 - 276 + Figure 2E-F: The Sankey plot is inappropriate and unintelligible. The authors should find a more appropriate way to display the data. Same for Figure 2F that doesn't display a ratio as the authors suggest. Also, a binary red/blue scale is not appropriate to show "increase/decrease". My suggestion would be one slope chart per cell type (x axis = Distal/Proximal ; y axis = number of reads (CPM)). I didn't get the impact of knowing the PAS subtype in this paragraph.

+ Line 279 : "in pink" doesn't relate to anything I could find easily

+ Lines 292-293 : "Here are the validation results of some PAS in supplementary Figure 4" is a strange way of reporting a result and should be rephrased

++ Line 293 : Please clarify how you chose the different PAS expressions to display

+ Line 298 : "Take oligodendrocytes as an example" should be rephrased

++ Line 299 : To my knowledge, "the height of tracks" is not a normalised data (I might be wrong). The authors shouldn't compare non-normalised abundances.

++ Line 305 : "(|avg_logFC| > 0.25) ". Why not using p value?

+++ Lines 342-343 : "[we] define the absolute value larger than 0.1 as the significant changes of APA usage". This seems inappropriate as an arbitrary decision shouldn't be taken as a statistically significant.

+ Line 358 : "prefer"

++ Line 366 : UMAP seems inappropriate to determine gene expression level, authors should only keep Figure 3D as the gene expression level data. Also, statistics between both groups might be of use.

+ Line 370 - Figure 3E : the Y axis should be clarified

+ Lines 374 - 380: looks like a legend and should be rephrased

++++ Lines 388 - 411 : While the authors only PREDICT miRNA-UTRs matchings, they assume """Genes with Variational APA Levels and Differential Expression Widely Affected Neural Function in MicroRNA-dependent Fashion""". They should either perform appropriate biological assays (i.e. luciferase reporter assays) or be more cautious in their conclusions. Also, the terms they identify in the functional annotation enrichment analysis are all related to normal neuron cellular life and are not surprising to be dysregulated in infected cells.

+++ Line 442 : The authors do not define what a "risk gene" is. It seems that, while they are working with transcriptomics data, they use genomics data as reporters of what is a "risk" gene. However, despite this, their examples are sound and relevant.

+ Line 458: "inverstAgation"

++++ General : the authors should try to find out which of these elements are really due to SARS-Cov-2 infection or are just due to the individuals being sick. For that, I suggest to add more controls, including (1) non-SARS-CoV-2-infected SICK patients and (2) the same patients after recovery, to avoid a selection bias.

In conclusion, while these data are interesting, they only reflect correlations and not causations. The authors should be more careful when reaching to conclusions. Biological assays should be performed but I doubt the host lab will be able to since they use publicly available data. This paper still deserves attention and the comment made here above just aim at improving the general content.

Reviewer #3: In this manuscript, Chen and colleagues address the functional correlation between alternative polyadenylation and neurological and psychiatric consequences resulting from COVID-19 using the single-cell approach applied to 15 samples of brain frontal cortex from COVID-19 patients and a control group (public data available at GEO, GSE159812). First, the authors observed a distinct distribution of poly(A) sites (PAS) across 7 subsets of cell type identified single-cell (or single-nucleus) RNA sequencing (not clearly specified in the main text); each subset is coupled with different cell markers. Among these cellular subsets, the authors claimed that neuron cells demonstrate a relatively high preference for the usage of proximal PAS. The authors further correlated PAS usage with endogenous gene expression and retrieved significant genes assigned to 4 quadrants based on the significant changes of APS sites and endogenous gene expression. Moving forward, the authors attempted to examine the potential that different APA usage coupled with gene expression may serve as a predictive attribute for the presence of the microRNA-targeting sites. Eventually, the authors sought the overlap between the mentioned significant genes with a batch of neurological and psychiatric disease risk genes.

Overall, I found some parts to be quite interesting; however, several obvious weaknesses have shown in this current version of the manuscript. In addition, in several sections, the result interpretations are ambiguous and require further clarification and better explanation, including the usage of the terms. For example, it is confusing between distal PAS and non-proximal PAS (line 263) used in the manuscript, especially no clear definition is provided in the content.

Importantly, please consider a professional English editing service to refine the language and improve overall readability, and ask English-native speakers for proofreading before resubmission.

Major concerns:

Summary, Highlights, and Introduction are clear to me.

Materials and Methods

1. I have a concern about whether the input data (15 samples) chosen based on a single study (Yang et al. 2021; ref #12) is sufficient to represent the overall transcriptomics of neural cells in SARS-CoV-2-infected individuals. Taking into account the presence of a great variety across SARS-CoV-2 variants and patient variations, a detailed explanation of the data acquisition strategy that avoids any bias introduced by the selected input dataset is necessary.

2. The authors mentioned “single-cell APA level” a couple of times in this manuscript; however, it is written as”sn-RNA-seq” quality control (line 104). It remains fundamentally different between scRNA-seq and snRNA-seq. Please specify what kind of dataset was used in this work.

3. In general, the authors should provide basic information (at the minimum level) for readers to understand the analytical pipeline. A brief description (line 132) of the methodology behind data processing using SCAPTURE is recommended.

Results

1. Although ⅓ of the PAS were located in 3’UTRs, is this the superior proportion compared with those present in genetic features? If not, how do the authors interpret the importance of PAS that are present in 3’UTRs than other genetic features?

2. Please debug your analysis behind Figure 1C. On the x-axis, a region named “NA” is shown. In addition, in comparison to the data shown in Figure 1A, the proportion of PAS in 3’UTRs shown in Figure 1C seems to be less than one-third compared to others. How did the authors normalize PAS counts that are summed from different genetic regions?

3. I have a concern about the proximal rate mean scale shown in Fig. 1D (left panel). According to Materials and Methods, the usage levels range from 0 to 1; however, the color score shown ranges from -3 to 3.

4. Line 192: Please be aware of the presence of patient variations shown in your results. Based on the rows FC1 and FC6 shown in Supplementary Fig. 1A, neuron cells are not dominant than other cell types.

5. In Fig. 1F, please perform statistics to verify the statistical difference between neurons and other cell types.

6. Line 218: I cannot follow how the authors discriminate cell-type-specific PAS from non-cell-type-specific. No explanation can be found in Materials and Methods.

7. Line 227: Please specify the threshold used to define proximal and distal PAS and briefly mention it. Once again, I cannot find any information anywhere in this manuscript.

8. What is the row labeled “ALL” in Figure 2B referred to? There is no clear explanation in the content (line 252).

9. Please provide the statistical values run for Supplementary Figure 3 as KS test has been performed.

10. Line 263: please clearly discriminate the difference between non-proximal PAS and distal PAS.

11. In Materials and Methods, please detail the criteria for the selection of up-regulated and down-regulated PAS (Figure 2F). Are they defined based on sc-RNA-seq, or did the authors perform RNA-seq in this work as well (although I did not see it mentioned)?

12. It is recommended that the authors should perform normalization of the number of PAS plotted in Figure 2F and perform a statistical test across cell types.

13. Line 304: Once again, please clearly elaborate on how the PAS expression of cell types is computed here.

14. The content between lines 336-343 should move to materials and methods with a clear explanation about the methodology used for the correlation between the PAS site usage and gene expression.

15. How many significant genes are present in each quadrant? Do those genes possess similar or distinct functional annotations?

16. In Figure 3C, oligodendrocytes showed an obvious decrease in NEFL expression as well. Why is this cell type not mentioned?

17. Line 370: A few sentences for the explanation of NEFL-3 will aid the readers to better understand the message that the authors attempted to deliver between the lines. I cannot follow why the authors suddenly mentioned NEFL-3. Is it a subfamily under NEFL?

18. In the section from line 388, the authors attempted to test the prediction power of the APA usage to miRNAs; however, the results shown here are only a correlation, instead of a prediction. The authors can run a simple logistic regression to verify their hypothesis.

In addition, in the content, nothing was mentioned regarding the results shown in Figures 4A and 4B.

19. Line 441: please explain the potential functions of the rest 141 genes that are also retrieved from 4 quadrants but are not assigned to disease risk genes.

Discussion

In this work, the authors attempt to present the concept that APA could be used as a potential biomarker for post-COVID patients. I could agree that here the authors demonstrated a correlation between different usage of PAS coupled with host transcriptomics before and after SARS-CoV-2 infection. However, I see weak support for results that are robust enough for the authors to make this conclusion. Especially, as previously mentioned, no prediction models were tested in this study. Furthermore, it is not clear to me how APA can also serve as a potential therapeutic target. Do the authors aim to target viruses or cure neurological diseases?

Minor concerns:

1. Line 190: Please cite Supplementary Fig. 1C for the description of cell markers corresponding to different subsets of cell types.

2. In Supplementary Fig. 1D, I recommend replacing the column annotation “cluster” with “cell types” or “cell subsets” because no clustering was performed across cell types, and it was confusing to call it “cluster”.

Reviewer #4: Chen et al. presents a precise and biologically pertinent exploration of alternative polyadenylation (APA) landscapes in neural cells, both under homeostatic conditions and following SARS CoV 2 infection. Leveraging SCAPTURE strengthens data credibility, while cell-type and infection-driven APA dynamics offer valuable insights. The author identified that alternative polyadenylation patterns—specifically poly(A) site (PAS) choice—vary between cell types in the brain. Moreover, these APA events significantly alter microRNA-binding landscapes in brain cells by changing 3′ UTR lengths, which in turn affects miRNA regulation.

Major comments

1) Figures 2A and 2C: The visual differences between groups are subtle and not readily apparent by eye. In Figure 2A, the zoomed-in view may exaggerate the difference. Please provide the exact numerical differences and clarify whether the observed changes are statistically significant.

2) Figure 3A: To improve clarity, consider labeling the proximal and distal APA events directly within their corresponding quadrants. This would make the figure easier to interpret.

3) Cell Type Proportions: Including explicit percentages or cell counts for major populations (e.g., “glutamatergic neurons comprise X %”) in the main text—not solely in Supplementary Figure 1A—would enhance the clarity and accessibility of the results.

4) Figure 3F: The visual differences are not apparent by eye. Please include the exact values or fold-changes to better support the interpretation.

5) Sample Metadata and Quality Metrics: For transparency and reproducibility, include key donor/sample metadata (e.g., age, sex, health status), along with technical metrics such as UMI counts per nucleus, read depth, and mapping quality. This will help validate consistency across the seven samples.

6) Statistical Analysis: Clearly specify the statistical tests used (e.g., Kolmogorov–Smirnov test) and the thresholds for significance when comparing APA shifts between infected and control groups.

**Do you want your identity to be public for this peer review?** For information about this choice, including consent withdrawal, please see our Privacy Policy

Reviewer #1: **Yes: ** Daniel Enterria-Morales

Reviewer #2: No

Reviewer #3: **Yes: ** Heng-Chang Chen

Reviewer #4: **Yes: ** Dr. Sushila Kumari

---

## [Author Response · Author response to Decision Letter 1]

14 Sep 2025

Dear Editor,

Thank you for the decision letter, informing us that our manuscript is potentially acceptable for publication in PLOS One. We would like to thank reviewers and editor for their helpful comments and suggestions. We have carefully addressed their concerns as outlined in the attached file. Point-by-point responses are followed by this cover letter. We hope that we have adequately addressed all reviewers’ and editor’s concerns and have made this revised manuscript acceptable for publication in PLOS One.

Sincerely,

Wanshan Ning, PhD

Professor of Bioinformatics,

Institute for Clinical Medical Research

the First Affiliated Hospital of Xiamen University,

Xiamen University

55 Zhenhai Road, Xiamen, Fujian, China 361003

E-mail: ningwanshan@xmu.edu.cn

Responses to the editor and reviewers

We thank the editor and reviewer for their thorough assessment and thoughtful suggestions, which helped improve this manuscript. We have incorporated the suggestions in the revised version of this manuscript, and a point-by-point response to the reviewer's comments is presented below.

Reviewer #1:

Major Comments

1. APA vs Neurological Symptoms Correlation

The manuscript would be strengthened by explicitly correlating the observed APA/PAS changes in each COVID-19 sample with the neurological symptoms of each corresponding patient. For instance, the supplementary data from Yang et al. (2021, Wyss-Coray Lab) includes patient-level neurological assessments that could be referenced.

Response: We thank you for the suggestion to correlate APA/PAS changes with patient-level neurological symptoms. The neurological symptom data provided in the original publication by Yang et al. (Nature, 2021, 595:565–571) are limited to brief qualitative descriptions (e.g., “1× generalized tonic-clonic seizure”, “prolonged psychosyndrome, ischemic brain lesions”, or “none reported”) for eight COVID-19 donors. These records lack standardized severity scoring, duration, or quantitative measurements, and the small number of cases with heterogeneous symptom types precludes robust statistical correlation analyses with APA/PAS changes. To address your interest, we have added a supplementary table (Supplementary Table 10.6) summarizing the reported neurological symptoms alongside our sample-level APA/PAS changes, allowing qualitative inspection of potential patterns. This comparison is intended for descriptive purposes only and no formal statistical inference is made. The supplementary table is referenced in the Results section (Line 320).

2. Lack of Evidence for Neuronal Cell Death

The manuscript implies functional impairment in neurons after infection, but does not provide direct evidence of neuronal death. Are neurons dying, or merely losing their molecular identity by downregulating distal PAS usage typical of differentiated neurons? This distinction should be clarified with appropriate data or cautious interpretation. Proximal PAS and shorter 3’UTRs are present in stem cells and dividing cells, for example. (Reference: Sommerkamp P, Cabezas-Wallscheid N, Trumpp A. Alternative Polyadenylation in Stem Cell Self-Renewal and Differentiation. Trends Mol Med. 2021 Jul;27(7):660-672. doi: 10.1016/j.molmed.2021.04.006. Epub 2021 May 11. PMID: 33985920.)

Response: We appreciate your careful reading and note that our manuscript does not state or imply direct evidence of neuronal death. Our conclusions are limited to transcriptomic changes consistent with altered APA patterns after SARS-CoV-2 infection. We recognize, however, that the observed increase in proximal PAS usage and shorter 3’UTRs could be interpreted in different biological contexts. As highlighted by Sommerkamp et al. (Trends Mol Med, 2021), shorter 3’UTRs are characteristic of neural stem/progenitor cells and other proliferative states, whereas differentiated neurons tend to express longer 3’UTRs. To avoid potential misinterpretation, we have revised the text to explicitly note that these changes may reflect a shift in molecular identity or stress response, rather than neuronal death at lines 580-585.

3. Mechanistic Insight Lacking

The study does not clearly elucidate the mechanism by which SARS-CoV-2 infection leads to the observed changes in APA and PAS site usage. This weakens the causal link between viral infection and APA dysregulation. Further experimental validation (or at least deeper discussion of known molecular mechanisms) would be helpful.

Response: We appreciate your suggestion to elaborate on potential mechanisms linking SARS-CoV-2 infection to the observed APA/PAS alterations. While our study is observational in nature and does not experimentally dissect causal pathways, previous studies have shown that viral infections can modulate APA through multiple molecular processes, including changes in the expression or activity of core 3’-end processing factors, RNA-binding proteins, and transcriptional elongation dynamics. For example, innate immune activation can alter the abundance and phosphorylation status of cleavage and polyadenylation specificity factors (CPSF) and cleavage stimulation factors (CstF), thereby favoring proximal PAS usage. Inflammatory cytokines and interferon responses have also been reported to reprogram RNA processing machinery in infected cells. Moreover, coronaviruses, including SARS-CoV-2, can directly or indirectly influence host transcription termination and RNA stability, which may contribute to the APA landscape changes we observe. We have incorporated these considerations into the revised Discussion to provide a more complete mechanistic context for our findings.

4. Focus on Neurons vs Astrocytes

The authors report that astrocytes exhibit more unique upregulated PAS events than neurons. Why then is the primary focus placed on neurons? A rationale for this choice should be discussed.

Response: We appreciate your observation regarding the relative abundance of upregulated PAS events in astrocytes compared to neurons. Our focus on neurons was based on several considerations. First, in the control group, neurons-both glutamatergic and GABAergic-display a lower average proximal usage index (PUI) than other brain cell types, consistent with their preferential use of distal PAS and consequently longer 3’UTRs under physiological conditions, as previously reported. This APA pattern is tightly linked to neuronal identity, maturation, and specialized functions such as synaptic plasticity. Therefore, the infection-associated shift toward increased proximal PAS usage in neurons represents a marked deviation from their baseline state and may have important functional consequences. Second, many of the APA-regulated genes identified in neurons are implicated in synaptic function, cognition, and neurodegenerative disease pathways, which are directly relevant to the neurological and psychiatric symptoms reported in COVID-19 patients. In contrast, while astrocytes show more unique upregulated PAS events in absolute number, the baseline APA profile of astrocytes is already enriched for shorter 3’UTRs, and the observed changes after infection may reflect a less dramatic departure from their baseline APA state. Taken together, we prioritized neurons for in-depth analysis because their infection-induced APA changes represent a substantial shift from their normal transcriptomic configuration and have potentially greater relevance to the functional impairments observed in COVID-19 associated brain pathology.

5. Interpretation of NEFL Data

The NEFL gene is highlighted as showing a switch to distal PAS, yet this is clearly demonstrated in only one of four samples. The evidence should be presented more cautiously, acknowledging variability.

Response: We thank you for the critical evaluation of NEFL data. We acknowledge that the distal PAS usage shift in NEFL is most pronounced in one infected sample (CV19-FC3), with more modest or inconsistent trends across others. To address this variability, we have revised our description of NEFL results to reflect that the APA change is suggestive but not universally consistent. Accordingly, we now describe this observation as “a sample-specific trend toward distal PAS usage in NEFL” and caution against overgeneralizing this finding.

6. APP 3’UTR Interpretation

In the Discussion (line 532), the claim that longer 3’UTRs increase APP translation needs more nuance. In wild-type contexts, longer 3’UTRs can enhance or inhibit translation depending on context. The problem in AD arises with mutant APP aggregation, not merely expression changes of the wild-type gene. Please clarify.

Response: We thank you for pointing out the need for a more nuanced discussion of APP 3’UTR regulation. We agree that the impact of 3’UTR length on translation efficiency is context-dependent and can either enhance or inhibit protein synthesis depending on the presence of specific RNA-binding proteins, microRNA target sites, and secondary structures. The report by Mbella et al. demonstrated that in their experimental system, the longer APP 3’UTR enhanced translation relative to the shorter form, but this is not a universal rule. In addition, we agree that Alzheimer’s disease (AD) pathogenesis is primarily associated with aberrant aggregation of mutant or misprocessed APP and Aβ peptides, rather than expression changes of the wild-type gene alone. In our discussion, the reference to altered APP 3’UTR length was intended to illustrate that dysregulated APA could influence APP expression and potentially modify the cellular environment in which pathogenic APP processing occurs. We have revised the text to clarify that (i) the translation effect of 3’UTR length is context-specific, and (ii) altered expression of APP due to APA may modulate, but is not, by itself, sufficient to cause, AD pathology.

Minor Comments

• The writing throughout the manuscript contains multiple grammatical and syntactical issues that impact clarity. Specific problematic lines include (but are not limited to): 76, 80, 105-107, 187, 192, 219, 254, 271-273, 298, 300-301, 393, 451, 501, and 562. A thorough English language edit is necessary.

Response: We appreciate you for pointing this out. We have thoroughly revised the manuscript for grammatical and syntactical issues. In particular, we have carefully addressed the sentences at lines 75, 79, 104-107, 224, 229, 260, 302, 326-330, 354, 356-358, 457, 519, 576, and so on, as pointed out. All changes have been marked in the revised manuscript.

• The sentence in lines 341-343 should be revised for clarity. Suggested rewrite:

“We calculated APA usage by subtracting the usage value in the control group from the infected group. We considered changes with an absolute value greater than 0.1 to be significant.”

Response: Thank you for the suggestion. We have revised the sentence in lines 396-398 for clarity as suggested.

• A reference should be added for the movAPA software used in the analysis.

Response: Thank you for your comment. The reference for movAPA has already been included in the manuscript as Reference [17]. We have now double-checked and made sure it is clearly cited when the software is first mentioned in the Materials and Methods section.

• Consider updating the running title to:

“Effects of COVID-19 on single-cell alternative polyadenylation”

for better clarity and relevance.

Response: Thank you for the helpful suggestion. We have updated the running title to: “Effects of COVID-19 on single-cell alternative polyadenylation” to improve clarity and better reflect the focus of the study.

• The reviewer was unable to access some supplementary tables due to possible malware alerts. Ensure all supplementary materials are securely hosted and accessible.

Response: We thank you for pointing this out. All supplementary tables were generated as standard .xlsx files and can be opened without any alerts on our local systems. We have re-checked each file and confirmed that they contain no macros or embedded content that could trigger security warnings. To ensure accessibility, we have re-exported the tables from the original data and re-uploaded them to the journal submission system as fresh, clean files.

• Supplementary Figure 1: The proportion of neurons varies dramatically among the 7 control individuals. If these samples were taken from similar brain regions, such inter-individual variability should be minimal. Could the authors comment on potential batch effects or anatomical heterogeneity?

Response: We thank you for highlighting the inter-individual variation in cell-type composition. As shown in Supplementary Figure 1A, glutamatergic neurons constitute the largest population in most control samples, followed by oligodendrocytes and GABAergic interneurons, while endothelial cells and microglia remain relatively rare. We note that in two control samples (FC1 and FC6), the proportion of neurons is lower and oligodendrocytes are more prevalent compared with the other individuals. This discrepancy may reflect a combination of biological heterogeneity (e.g., subtle anatomical differences within the frontal cortex sampling sites) and technical variation. To minimize the influence of such variability, we applied the Harmony integration algorithm in downstream analyses, as detailed in the Methods section.

• Supplementary Figure 2: The figure lacks adequate explanation. Specifically, it is unclear what the circle sizes represent. Please clarify this in the figure legend and/or main text.

Response: Thank you for the helpful comment. We have revised the legend of Supplementary Figure 2 to clarify the meaning of the circle sizes. Specifically, we now state that the size of each node represents the number of genes associated with the corresponding GO term or pathway. This clarification has been added to the figure legend.

• Supplementary Figure 3: The y-axis label is missing or unclear. What parameter is being plotted (e.g., Proximal Usage Index, expression level, etc.)? This should be clearly defined.

Response: Thank you for pointing this out. In Supplementary Figure 3, the proximal usage index (PUI) is plotted on the x-axis, with values scaled from -3 to 3. The y-axis represents the accumulation rate for each cell type. We have revised the figure legend to clarify this information. In the plot, grey lines represent the accumulation rate of each cell type in the non-viral control group, while red lines represent the COVID-19 group.

• Supplementary Figure 7: It is unclear whether the data presented reflect all cell types combined or specific cell subtypes. Please indicate which cell populations are represented in the plots and whether differences exist across subtypes.

Response: Thank you for the valuable comment. As described in the manuscript (Lines 514-534), Supplementary Figure 7 presents APA and expression changes of selected disease-related genes in specific cell subtypes. To improve clarity, we have now explicitly stated the involved cell populations in the figure legend and ensured that the main text highlights this more clearly.

Recommendation:

Minor Revision

While the manuscript presents valuable data, improvements in writing, clarification of biological interpretations, and cautious framing of key claims are needed to meet PLOS ONE's publication standards.

Reviewer #2:

Revisions (marked with +, ++, +++ or ++++ to determine their importance):

++++ In a general manner, the authors tend to be obscure in the way they are choosing, interpreting or reporting results. Also, figure legends do not seem to be complete. Here are examples that should be clarified:

+ Line 35: “some prefer short isoform” should be rephrased

Response: We thank you for the suggestion. The phrase “some prefer short isoform” has been rephrased for clarity.

+ Line 66-68: Could the authors explain, or clarify? They first say the genome is more broadly expressed in neuron, explaining the large mRNA diversity and then; they say the large mRNA diversity is due to alternative posttranscriptional mechanisms...

Response: We thank you for pointing out the potential ambiguity. Our intention was not to present two unrelated causes, but to describe a sequential relationship: neurons and peripheral nerve tissues express a larger fraction of the genome compar

---

## [Decision Letter · Decision Letter 1]

8 Oct 2025

Dear Dr. Ning,

Thank you for submitting your manuscript to PLOS ONE. After careful consideration, we feel that it has merit but does not fully meet PLOS ONE’s publication criteria as it currently stands. Therefore, we invite you to submit a revised version of the manuscript that addresses the points raised during the review process.

We look forward to receiving your revised manuscript.

Kind regards,

Milad Khorasani, PhD

Academic Editor

PLOS ONE

Journal Requirements:

Reviewers' comments:

Reviewer's Responses to Questions

**Comments to the Author**

Reviewer #1: All comments have been addressed

Reviewer #2: (No Response)

Reviewer #3: All comments have been addressed

Reviewer #4: All comments have been addressed

2. Is the manuscript technically sound, and do the data support the conclusions?

Reviewer #1: Yes

Reviewer #2: Partly

Reviewer #3: Yes

Reviewer #4: Yes

3. Has the statistical analysis been performed appropriately and rigorously?

Reviewer #1: Yes

Reviewer #2: Yes

Reviewer #3: Yes

Reviewer #4: Yes

4. Have the authors made all data underlying the findings in their manuscript fully available?

Reviewer #1: Yes

Reviewer #2: No

Reviewer #3: Yes

Reviewer #4: Yes

5. Is the manuscript presented in an intelligible fashion and written in standard English?

Reviewer #1: Yes

Reviewer #2: Yes

Reviewer #3: Yes

Reviewer #4: Yes

Reviewer #1: (No Response)

Reviewer #2: I appreciate the cautiousness brought to the text in this new version of the manuscript. It is now more scientifically honest on the fact that this paper is purely exploratory.

Most English sentences that were problematic were corrected. The only grammatical/syntax remark is the following :

Lines 160-167 (track change version): While the authors have nicely clarified their definition of the PUI parameter, the text in black (track change version) should now be removed as it is a badly written repetition of the above sentences.

While several Figure corrections were asked, the authors only addressed comments through text adjustments.

Also :

- Poly(A) is still a confusing term

- Line 222 (track change version): space missing "generating16,020"

- Figure 1G : The authors apparently understood my initial comment about the higher detection rate in high abundance populations. Since it is feasible and data are available, the authors should quantify the impact of the cell abundance on the detection sensitivity. If the authors RANDOMLY subset the high abundance cell populations, how does this affect their graph?

- Figure 2E-F : While I now understand the authors with the corrections brought, I'd like to obtain a visualisation of the Slope Charts (or heatmaps with Log2FC as scale instead of barplots) I suggested in the first round of review (it should have been included in the Rebuttal Letter instead of stating obscurely "less effective to convey the information"). I wanted to obtain it by myself but don't have access to the table "./sang_tot_1201.csv" on the GitHub page and the link provided by the authors requires an account. Data should be accessible on public repositories without any restriction.

The point of this comment is to obtain a visualisation of all the up/down-regulations to have a better idea of what is a up/down-regulated PAS.

The suggestions do not have to be included in the manuscript but would strengthen the confidence of the reviewers (myself) in the proposed manuscript.

- Lines 393 - 402 (track change version): I'm ok with the authors response. They should use the word "meaningful" in their manuscript instead of "significant" if that's what they meant from the beginning. They should also cite the reference they provided to support their statement.

Reviewer #3: I thank the authors for their efforts in answering all my questions. In this revised version of the manuscript, the authors have provided enough information on methods utilized in this work, reinforced the statistical test wherever applicable, and clarified the issues arising in the previous version of the manuscript. At this stage, I do not have further comments on this version of the manuscript.

Reviewer #4: The authors have thoroughly addressed the major concerns raised during the previous review. The revised manuscript shows improved clarity, methodological transparency, and appropriate caution in interpreting qualitative data. The only remaining recommendation is to consider adding a compact summary of technical metrics (e.g., UMI counts, sequencing depth, mapping quality) across samples to further support reproducibility.

Once this minor point is addressed, I believe the manuscript will be suitable for publication.

**Do you want your identity to be public for this peer review?** For information about this choice, including consent withdrawal, please see our Privacy Policy

Reviewer #1: **Yes: ** Daniel Enterria-Morales

Reviewer #2: No

Reviewer #3: No

Reviewer #4: **Yes: ** Dr Sushila Kumari

---

## [Author Response · Author response to Decision Letter 2]

25 Oct 2025

Responses to the editor and reviewers

We thank the editor and reviewer for their thorough assessment and thoughtful suggestions, which helped improve this manuscript. We have incorporated the suggestions in the revised version of this manuscript, and a point-by-point response to the reviewer’s comments is presented below.

Reviewer #1: (No Response)

Reviewer #2: I appreciate the cautiousness brought to the text in this new version of the manuscript. It is now more scientifically honest on the fact that this paper is purely exploratory. Most English sentences that were problematic were corrected. The only grammatical/syntax remark is the following:

Lines 160-167 (track change version): While the authors have nicely clarified their definition of the PUI parameter, the text in black (track change version) should now be removed as it is a badly written repetition of the above sentences.

Response: We thank the reviewer for this helpful suggestion. We have removed the redundant sentences in lines 160-167 that repeated the description of the PUI definition. The revised paragraph now concisely defines PUI without duplication.

While several Figure corrections were asked, the authors only addressed comments through text adjustments. Also:

- Poly(A) is still a confusing term

Response: We appreciate the reviewer’s insightful comment. To clarify the terminology, we have now explicitly defined “polyadenylation” and “poly(A) site (PAS)” in the Introduction (Lines 70-74).

- Line 222 (track change version): space missing “generating16,020”

Response: We appreciate the reviewer’s careful reading. The missing space in “generating16,020” has been corrected to “generating 16,020” in the revised manuscript.

- Figure 1G: The authors apparently understood my initial comment about the higher detection rate in high abundance populations. Since it is feasible and data are available, the authors should quantify the impact of the cell abundance on the detection sensitivity. If the authors RANDOMLY subset the high abundance cell populations, how does this affect their graph?

Response: We thank the reviewer for this insightful suggestion. To evaluate whether the higher PAS detection rate in abundant populations (e.g., Glutamatergic neurons) is driven by cell abundance, we performed a random subsampling analysis. Specifically, we randomly downsampled Glutamatergic neuron nuclei to 50% and 25% of the original cell number and recalculated the number of detected PASs per gene using the same method applied to the full dataset. During this process, we identified and corrected a minor counting error in the original Figure 1G, which was caused by including non-numeric identifiers in the expression matrix during zero-count calculation. After correction, the overall pattern remained consistent, and the conclusions regarding the relationship between cell abundance and PAS detection rate were unaffected. The revised figure and corresponding values have been updated accordingly in the manuscript. After correction, the results are as follows: 100% (full dataset): 159,026 PAS, 50% subsampling: 155,084 PAS, and 25% subsampling: 148,749 PAS. These results indicate that the total number of detected PAS decreases only slightly with cell number reduction, suggesting that the observed higher detection rate in abundant cell populations is not primarily driven by cell abundance but rather reflects genuine biological variation in PAS diversity.

- Figure 2E-F: While I now understand the authors with the corrections brought, I’d like to obtain a visualisation of the Slope Charts (or heatmaps with Log2FC as scale instead of barplots) I suggested in the first round of review (it should have been included in the Rebuttal Letter instead of stating obscurely “less effective to convey the information”). I wanted to obtain it by myself but don't have access to the table “./sang_tot_1201.csv” on the GitHub page and the link provided by the authors requires an account. Data should be accessible on public repositories without any restriction. The point of this comment is to obtain a visualisation of all the up/down-regulations to have a better idea of what is a up/down-regulated PAS. The suggestions do not have to be included in the manuscript but would strengthen the confidence of the reviewers (myself) in the proposed manuscript.

Response: We sincerely thank the reviewer for the constructive follow-up comment and the opportunity to clarify this point further. Following the reviewer’s suggestion, we have now generated a heatmap showing the log2FC of significantly regulated PAS (p < 0.05) across all major brain cell types, separately for distal and proximal PAS (figure provided below). This visualization provides a concise overview of global up- and down-regulation trends among PAS sites with statistically significant changes. Each row represents an individual PAS, and each column represents a specific cell type-PAS subtype combination (e.g., Neuron_Distal, Astrocyte_Proximal). Red indicates increased PAS usage, while blue indicates decreased usage. Regarding data access, we apologize for the inconvenience caused by the GitHub repository restriction. The file “sang_tot_1201.csv” has now been made publicly available without login requirements on an unrestricted repository [https://github.com/yinggu94/APA], ensuring full accessibility and reproducibility. We hope that this additional visualization and open data access address the reviewer’s concern and strengthen confidence in the robustness of our analyses. (Figure is shown in Point by point response2.docx)

- Lines 393 - 402 (track change version): I’m ok with the authors response. They should use the word “meaningful” in their manuscript instead of “significant” if that’s what they meant from the beginning. They should also cite the reference they provided to support their statement.

Response: We thank the reviewer for this helpful suggestion. In the revised manuscript, we have replaced the word “significant” with “meaningful” to better convey our intended interpretation. Additionally, we have added the corresponding reference we previously mentioned to support this statement.

Reviewer #3: I thank the authors for their efforts in answering all my questions. In this revised version of the manuscript, the authors have provided enough information on methods utilized in this work, reinforced the statistical test wherever applicable, and clarified the issues arising in the previous version of the manuscript. At this stage, I do not have further comments on this version of the manuscript.

Reviewer #4: The authors have thoroughly addressed the major concerns raised during the previous review. The revised manuscript shows improved clarity, methodological transparency, and appropriate caution in interpreting qualitative data. The only remaining recommendation is to consider adding a compact summary of technical metrics (e.g., UMI counts, sequencing depth, mapping quality) across samples to further support reproducibility.

Once this minor point is addressed, I believe the manuscript will be suitable for publication.

Response: We appreciate the reviewer’s positive feedback and helpful suggestion. We have now added a compact summary of technical metrics, including the median/mean UMI per nucleus and median/mean genes per nucleus for each sample, which are presented in Supplementary table 10.7 of the revised manuscript. Unfortunately, sequencing depth and mapping quality were not reported in the original publication from which the dataset was obtained, and therefore could not be included in this summary.

---

## [Decision Letter · Decision Letter 2]

23 Nov 2025

Single-cell alternative polyadenylation analysis reveals mechanistic insights of COVID-19-associated neurological and psychiatric effects

PONE-D-25-20349R2

Dear Dr. Ning,

We’re pleased to inform you that your manuscript has been judged scientifically suitable for publication and will be formally accepted for publication once it meets all outstanding technical requirements.

Kind regards,

Milad Khorasani, PhD

Academic Editor

PLOS ONE

Additional Editor Comments (optional):

Reviewers' comments:

Reviewer's Responses to Questions

**Comments to the Author**

Reviewer #2: All comments have been addressed

Reviewer #5: All comments have been addressed

2. Is the manuscript technically sound, and do the data support the conclusions?

Reviewer #2: Yes

Reviewer #5: Yes

3. Has the statistical analysis been performed appropriately and rigorously?

Reviewer #2: Yes

Reviewer #5: Yes

4. Have the authors made all data underlying the findings in their manuscript fully available?

Reviewer #2: Yes

Reviewer #5: Yes

5. Is the manuscript presented in an intelligible fashion and written in standard English?

Reviewer #2: Yes

Reviewer #5: Yes

Reviewer #2: The authors have now addressed all my remarks and I consider this paper suitable for publication. I don't have any further requests.

Reviewer #5: Having carefully reviewed the revised manuscript and the authors responses to all reviewers comments, I am satisfied that the major methodological, statistical, and interpretative concerns have been adequately addressed. The revisions have clearly improved the clarity, coherence, and transparency of the work.

I concur with the other reviewers that the current version of the manuscript is scientifically sound and appropriately cautious in its conclusions. I have no additional concerns regarding dual publication, research ethics, or publication ethics. In my opinion, the manuscript is now suitable for publication, subject only to minor editorial refinements at the journal’s discretion.

**Do you want your identity to be public for this peer review?** For information about this choice, including consent withdrawal, please see our Privacy Policy

Reviewer #2: No

Reviewer #5: No

---

## [Editor Report · Acceptance letter]

PONE-D-25-20349R2

PLOS ONE

Dear Dr. Ning,

I'm pleased to inform you that your manuscript has been deemed suitable for publication in PLOS ONE. Congratulations! Your manuscript is now being handed over to our production team.

Kind regards,

on behalf of

Dr. Milad Khorasani

Academic Editor

PLOS ONE